# What Can the Neural Tangent Kernel Tell Us About Adversarial Robustness?

**Nikolaos Tsilivis**
Center for Data Science
New York University
nt2231@nyu.edu

**Julia Kempe**
Center for Data Science and
Courant Institute of Mathematical Sciences
New York University
kempe@nyu.edu

## Abstract

The adversarial vulnerability of neural nets, and subsequent techniques to create robust models have attracted significant attention; yet we still lack a full understanding of this phenomenon. Here, we study adversarial examples of trained neural networks through analytical tools afforded by recent theory advances connecting neural networks and kernel methods, namely the Neural Tangent Kernel (NTK), following a growing body of work that leverages the NTK approximation to successfully analyze important deep learning phenomena and design algorithms for new applications. We show how NTKs allow to generate adversarial examples in a "training-free" fashion, and demonstrate that they transfer to fool their finite-width neural net counterparts in the "lazy" regime. We leverage this connection to provide an alternative view on robust and non-robust features, which have been suggested to underlie the adversarial brittleness of neural nets. Specifically, we define and study features induced by the eigendecomposition of the kernel to better understand the role of robust and non-robust features, the reliance on both for standard classification and the robustness-accuracy trade-off. We find that such features are surprisingly consistent across architectures, and that robust features tend to correspond to the largest eigenvalues of the model, and thus are learned early during training. Our framework allows us to identify and visualize non-robust yet useful features. Finally, we shed light on the robustness mechanism underlying adversarial training of neural nets used in practice: quantifying the evolution of the associated empirical NTK, we demonstrate that its dynamics falls much earlier into the "lazy" regime and manifests a much stronger form of the well known bias to prioritize learning features within the top eigenspaces of the kernel, compared to standard training.

## 1 Introduction

Despite the tremendous success of deep neural networks in many computer vision and language modeling tasks, as well as in scientific discoveries, their properties and the reasons for their success are still poorly understood. Focusing on computer vision, a particularly surprising phenomenon evidencing that those machines drift away from how humans perform image recognition is the presence of *adversarial examples*, images that are almost identical to the original ones, yet are misclassified by otherwise accurate models.

Since their discovery (Szegedy et al., 2014), a vast amount of work has been devoted to understanding the sources of adversarial examples and explanations include, but are not limited to, the close to linear operating mode of neural nets (Goodfellow et al., 2015), the curse of dimensionality carried by the input space (Goodfellow et al., 2015, Simon-Gabriel et al., 2019), insufficient model capacity (Tsipras et al., 2019, Nakkiran, 2019) or spurious correlations found in common datasets (Ilyas et al., 2019).

36th Conference on Neural Information Processing Systems (NeurIPS 2022).

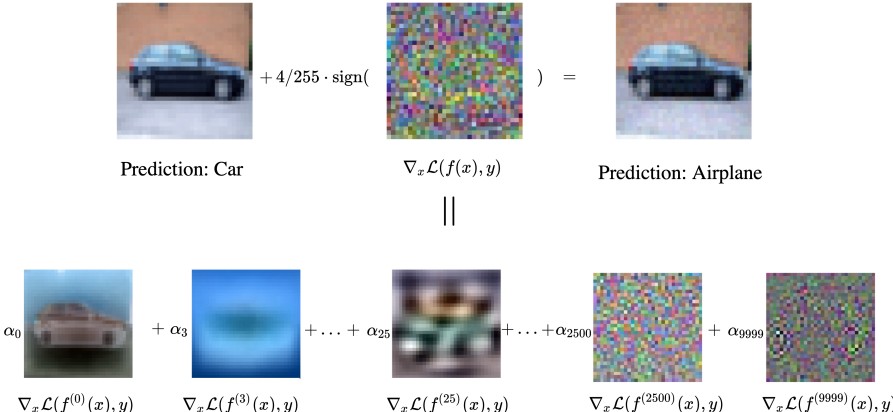

Figure 1: **Top**. Standard setup of an adversarial attack, where a barely perceivable perturbation is added to an image to confuse an accurate classifier. **Bottom**. The correspondence between neural networks and kernel machines allows to visualize a decomposition of this perturbation, each part attributed to a different feature of the model. The first few features tend to be *robust*.

In particular, one widespread viewpoint is that adversarial vulnerability is the result of a model's sensitivity to imperceptible yet well-generalizing features in the data, so called *useful non-robust* features, giving rise to a trade-off between accuracy and robustness (Tsipras et al., 2019, Zhang et al., 2019). This gradual understanding has enabled the design of training algorithms, that provide convincing, yet partial, remedies to the problem; the most prominent of them being adversarial training and its many variants (Goodfellow et al., 2015, Madry et al., 2018, Croce et al., 2021). Yet we are far from a mature, unified theory of robustness that is powerful enough to universally guide engineering choices or defense mechanisms.

In this work, we aim to get a deeper understanding of adversarial robustness (or lack thereof) by focusing on the recently established connection of neural networks with kernel machines. Infinitely wide neural networks, trained via gradient descent with infinitesimal learning rate, provably become kernel machines with a data-independent, but architecture dependent kernel - its Neural Tangent Kernel (NTK) - that remains constant during training (Jacot et al., 2018, Lee et al., 2019, Arora et al., 2019b, Liu et al., 2020). The analytical tools afforded by the rich theory of kernels have resulted in progress in understanding the optimization landscape and generalization capabilities of neural networks (Du et al., 2019, Arora et al., 2019a), together with the discovery of interesting deep learning phenomena (Fort et al., 2020, Ortiz-Jiménez et al., 2021), while also inspiring practical advances in diverse areas of applications such as the design of better classifiers (Shankar et al., 2020), efficient neural architecture search (Chen et al., 2021), low-dimensional tasks in graphics (Tancik et al., 2020) and dataset distillation (Nguyen et al., 2021). While the NTK approximation is increasingly utllized, even for finite width neural nets, little is known about the adversarial robustness properties of these infinitely wide models.

**Our contribution:** Our work inscribes itself into the quest to leverage analytical tools afforded by kernel methods, in particular spectral analysis, to track properties of interest in the associated neural nets, in this case as they pertain to robustness. To this end, we first demonstrate that adversarial perturbations generated *analytically* with the NTK can successfully lead the associated trained wide neural networks (in the kernel-regime) to misclassify, thus allowing kernels to faithfully predict the lack of robustness of those trained neural networks. In other words, adversarial (non-) robustness transfers from kernels to networks; and adversarial perturbations generated via kernels resemble those generated by the corresponding trained networks. One implication of this transferability is that we can analytically devise adversarial examples that do not require access to the trained model and in particular its weights; instead these "blind spots" may be calculated a-priori, before training starts.

A perhaps even more crucial implication of the NTK approach to robustness relates to the *understanding* of adversarial examples. Indeed, we show how the spectrum of the NTK provides an alternative way to define *features* of the model, to classify them according to their robustness and usefulness for

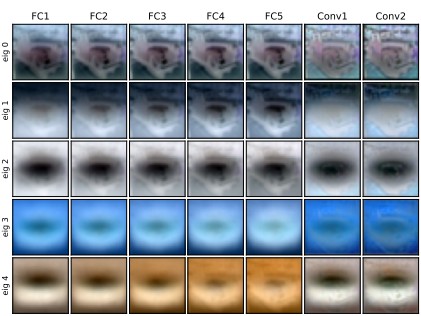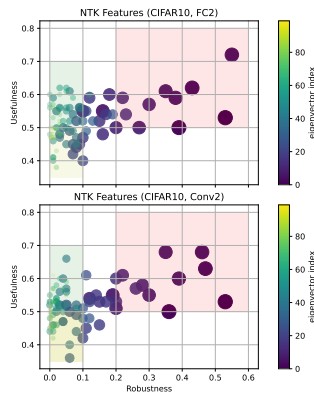

Figure 2: **Left**: Top 5 features for 7 different kernel architectures for a car image extracted from the CIFAR10 dataset when trained on car and plane images. **Right**: Features according to their robustness (x-axis) and usefulness (y-axis). Larger/darker bullets correspond to larger eigenvalues. *Useful* features have $> 0.5$-usefulness; shaded boxes are meant to help visualize useful-robust regions.

correct predictions and visually inspect them via their contribution to the adversarial perturbation (see Fig. 1). This in turn allows us to verify previously conjectured properties of standard classifiers; dependence on both *robust* and *non-robust* features in the data (Tsipras et al., 2019), and tradeoff of accuracy and robustness during training. In particular we observe that features tend to be rather invariable across architectures, and that robust features tend to correspond to the *top* of the eigenspectrum (see Fig. 2), and as such are learned first by the corresponding wide nets (Arora et al., 2019a, Jacot et al., 2018). Moreover, we are able to visualize useful non-robust features of standard models (Fig. 4). While this conceptual feature distinction has been highly influential in recent works that study the robustness of deep neural networks (see for example (Allen-Zhu and Li, 2022, Kim et al., 2021, Springer et al., 2021)), to the best of our knowledge, none of them has explicitly demonstrated the dependence of networks on such feature functions (except for simple linear models (Goh, 2019)). Rather, these works either reveal such features in some indirect fashion, or accept their existence as an assumption. Here, we show that Neural Tangent Kernel theory endows us with a natural definition of features through its eigen-decomposition and provides a way to *visualise and inspect robust and non-robust features directly* on the function space of trained neural networks.

Interestingly, this connection also enables us to empirically demonstrate that robust features of standard models alone are not enough for robust classification. Aiming to understand, then, what makes robust models robust, we track the *evolution* of the data-dependent *empirical* NTK during *adversarial training* of neural networks used in practice. Prior experimental work has found that networks with non-trivial width to depth ratio which are trained with large learning rates, depart from the NTK regime and fall in the so-called "rich feature" regime, where the NTK changes substantially during training (Geiger et al., 2020, Fort et al., 2020, Baratin et al., 2021, Ortiz-Jiménez et al., 2021). In our work, which to the best of our knowledge is the first to provide insights on how the kernel behaves during adversarial training, we find that the NTK evolves much faster compared to standard training, simultaneously both changing its features and assigning more importance to the more robust ones, giving direct insight into the mechanism at play during adversarial training (see Fig. 6). In summary, the contributions of our work are the following:

- We discuss how to generate adversarial examples for infinitely-wide neural networks via the NTK, and show that they transfer to fool their associated (finite width) nets in the appropriate regime, yielding a "training-free" attack without need to access model weights (Sec. 3).

- Using the spectrum of the NTK, we give an alternative definition of features, providing a natural decomposition or perturbations into robust and non-robust parts (Tsipras et al., 2019, Ilyas et al., 2019) (Fig. 1). We confirm that robust features overwhelmingly correspond to the top part of the eigenspectrum; hence they are learned early on in training. We bolster previously conjectured hypotheses that prediction relies on both robust and non-robust features and that robustness is traded for accuracy during standard training. Further, we

show that only utilizing the robust features of standard models is not sufficient for robust classification (Sec. 4).

- We turn to finite-width neural nets with standard parameters to study the *dynamics* of their empirical NTK during *adversarial training*. We show that the kernel rotates in a way that enables both new (robust) feature learning and that drastically increases of the importance (relative weight) of the robust features over the non-robust ones. We further highlight the structural differences of the kernel change during adversarial training versus standard training and observe that the kernel seems to enter the "lazy" regime much faster (Sec. 5).

Collectively, our findings may help explain many phenomena present in the adversarial ML literature and further elucidate both the vulnerability of standard models and the robustness of adversarially trained ones. We provide code to visualize features induced by kernels, giving a unique and principled way to inspect features induced by standardly trained nets (available at `https://github.com/Tsili42/adv-ntk`).

**Related work:** To the best of our knowledge the only prior work that leverages NTK theory to derive perturbations in some adversarial setting is due to Yuan and Wu (2021), yet with entirely different focus. It deals with what is coined *generalization attacks*: the process of altering the training data distribution to prevent models to generalise on clean data. Bai et al. (2021) study aspects of robust models through their linearized sub-networks, but do not leverage NTKs.

## 2 Preliminaries

We introduce background material and definitions important to our analysis. Here, we restrict ourselves to binary classification, to keep notation light. We defer the multiclass case, complete definitions and a more detailed discussion of prior work to the Appendix.

### 2.1 Adversarial Examples

Let $f$ be a classifier, $\mathbf{x}$ be an input (e.g. a natural image) and $y$ its label (e.g. the image class). Then, given that $f$ is an accurate classifier on $\mathbf{x}$, $\tilde{\mathbf{x}}$ is an adversarial example (Szegedy et al., 2014) for $f$ if

- i) the distance $d(\mathbf{x}, \tilde{\mathbf{x}})$ is small. Common choices in computer vision are the $\ell_p$ norms, especially the $\ell_\infty$ norm on which we focus henceforth, and
- ii) $f(\tilde{\mathbf{x}}) \neq y$. That is, the perturbed input is being misclassified.

Given a loss function $\mathcal{L}$, such as cross-entropy, one can construct an adversarial example $\tilde{\mathbf{x}} = \mathbf{x} + \boldsymbol{\eta}$ by finding the perturbation $\boldsymbol{\eta}$ that produces the maximal increase of the loss, solving

$$\boldsymbol{\eta} = \arg \max_{\|\boldsymbol{\eta}\|_\infty \leq \epsilon} \mathcal{L}(f(\mathbf{x} + \boldsymbol{\eta}), y), \tag{1}$$

for some $\epsilon > 0$ that quantifies the dissimilarity between the two examples. In general, this is a non-convex problem and one can resort to first order methods (Goodfellow et al., 2015)

$$\tilde{\mathbf{x}} = \mathbf{x} + \epsilon \cdot \text{sign}\left(\nabla_{\mathbf{x}} \mathcal{L}(f(\mathbf{x}), y)\right), \tag{2}$$

or iterative versions for solving it (Kurakin et al., 2017, Madry et al., 2018). The former method is usually called *Fast Gradient Sign Method (FGSM)* and the latter *Projected Gradient Descent (PGD)*. These methods are able to produce examples that are being misclassified by common neural networks with a probability that approaches 1 (Carlini and Wagner, 2017). Even more surprisingly, it has been observed that adversarial examples crafted to "fool" one machine learning model are consistently capable of "fooling" others (Papernot et al., 2016, 2017), a phenomenon that is known as the *transferability* of adversarial examples. Finally, *adversarial training* refers to the alteration of the training procedure to include adversarial samples for teaching the model to be robust (Goodfellow et al., 2015, Madry et al., 2018) and empirically holds as the strongest defense against adversarial examples (Madry et al., 2018, Zhang et al., 2019).

### 2.2 Robust and Non-Robust Features

Despite a vast amount of research, the reasons behind the existence of adversarial examples are not perfectly clear. A line of work has argued that a central reason is the presence of robust and

non-robust features in the data that standard models learn to rely upon (Tsipras et al., 2019, Ilyas et al., 2019). In particular it is conjectured that reliance on *useful but non-robust* features during training is responsible for the brittleness of neural nets. Here, we slightly adapt the feature definitions of (Ilyas et al., 2019)[1], and extend them to multi-class problems (see Appendix A).

Let $\mathcal{D}$ be the data generating distribution with $x \in \mathcal{X}$ and $y \in \{\pm 1\}$. We define a *feature* as a function $\phi : \mathcal{X} \to \mathbb{R}$ and distinguish how they perform as classifiers. Fix $\rho, \gamma \geq 0$:

1. $\rho$-**Useful** feature: A feature $\phi$ is called $\rho$-*useful* if

$$\mathbb{E}_{x,y \sim \mathcal{D}}\left[\mathbb{1}_{\{\text{sign}[\phi(x)]=y\}}\right] = \rho \tag{3}$$

2. $\gamma$-**Robust** feature: A feature $\phi$ is called $\gamma$-*robust* if it remains useful under any perturbation inside a bounded "ball" $\mathcal{B}$, that is if

$$\mathbb{E}_{x,y \sim \mathcal{D}}\left[\inf_{\delta \in \mathcal{B}} \mathbb{1}_{\{\text{sign}[\phi(x+\delta)]=y\}}\right] = \gamma \tag{4}$$

In general, a feature adds predictive value if it gives an advantage above guessing the most likely label, i.e. $\rho > \max_{y' \in \{\pm 1\}} \mathbb{E}_{x,y \sim \mathcal{D}}[\mathbb{1}_{\{y'=y\}}]$, and we will speak of "useful" features in this case, omitting the $\rho$. We will call such a feature **useful, non-robust** if it is useful, but $\gamma$-robust only for $\gamma = 0$ or very close to 0, depending on context.

The vast majority of works imagines features as being induced by the *activations* of neurons in the net, most commonly those of the penultimate layer (*representation-layer* features), but the previous formal definitions are in no way restricted to activations, and we will show how to exploit them using the eigenspectrum of the NTK. In particular, in Sec. 4, we demonstrate that the above framework agrees perfectly with features induced by the eigenspectrum of the NTK of a network, providing a natural way to decompose the predictions of the NTK into such feature functions. In particular we can identify robust, useful, and, indeed, useful non-robust features.

## 2.3 Neural Tangent Kernel

Let $f : \mathbb{R}^d \to \mathbb{R}$ be a (scalar) neural network with a linear final layer parameterized by a set of weights $\mathbf{w} \in \mathbb{R}^p$ and $\{\mathcal{X}, \mathcal{Y}\}$ be a dataset of size $n$, with $\mathcal{X} \in \mathbb{R}^{n \times d}$ and $\mathcal{Y} \in \{\pm 1\}^n$. Linearized training methods study the first order approximation

$$f(\mathbf{x}; \mathbf{w}_{t+1}) = f(\mathbf{x}; \mathbf{w}_t) + \nabla_{\mathbf{w}} f(\mathbf{x}; \mathbf{w}_t)^\top (\mathbf{w}_{t+1} - \mathbf{w}_t). \tag{5}$$

The network gradient $\nabla_{\mathbf{w}} f(\mathbf{x}; \mathbf{w}_t)$ induces a kernel function $\Theta_t : \mathbb{R}^d \times \mathbb{R}^d \to \mathbb{R}$, usually referred as the *Neural Tangent Kernel (NTK)* of the model

$$\Theta_t(\mathbf{x}, \mathbf{x}') = \nabla_{\mathbf{w}} f(\mathbf{x}; \mathbf{w}_t)^\top \nabla_{\mathbf{w}} f(\mathbf{x}'; \mathbf{w}_t). \tag{6}$$

This kernel describes the dynamics with infinitesimal learning rate (gradient flow). In general, the tangent space spanned by the $\nabla_{\mathbf{w}} f(\mathbf{x}; \mathbf{w}_t)$ twists substantially during training, and learning with the Gram matrix of Eq. (6) (empirical NTK) corresponds to training along an intermediate tangent plane. Remarkably, however, in the infinite width limit with appropriate initialization and low learning rate, it has been shown that $f$ becomes a *linear* function of the parameters (Jacot et al., 2018, Liu et al., 2020), and the NTK remains *constant* ($\Theta_t = \Theta_0 =: \Theta$). Then, for learning with $\ell_2$ loss the training dynamics of infinitely wide networks admits a closed form solution corresponding to kernel regression (Jacot et al., 2018, Lee et al., 2019, Arora et al., 2019b)

$$f_t(\mathbf{x}) = \Theta(\mathbf{x}, \mathcal{X})^\top \Theta^{-1}(\mathcal{X}, \mathcal{X})(I - e^{-\lambda \Theta(\mathcal{X}, \mathcal{X})t})\mathcal{Y}, \tag{7}$$

where $\mathbf{x} \in \mathbb{R}^d$ is any input (training or testing), $t$ denotes the time evolution of gradient descent, $\lambda$ is the (small) learning rate and, slightly abusing notation, $\Theta(\mathcal{X}, \mathcal{X}) \in \mathbb{R}^{n \times n}$ denotes the matrix containing the pairwise training values of the NTK, $\Theta(\mathcal{X}, \mathcal{X})_{ij} = \Theta(\mathbf{x}_i, \mathbf{x}_j)$, and similarly for $\Theta(\mathbf{x}, \mathcal{X}) \in \mathbb{R}^n$. To be precise, Eq. (7) gives the *mean* output of the network using a weight-independent kernel with variance depending on the initialization[2].

---

[1] We distinguish useful and robust features based on their accuracy as classifiers, not in terms of correlation with the labels as in Ilyas et al. (2019), allowing a natural extension to the multi-class setting. For robustness, we consider any accuracy bounded away from zero as robust, quantifying that an adversary cannot drive accuracy to zero entirely.

[2] For that reason, in the experiments, we often compare this with the centered prediction of the actual neural network, $f - f_0$, as is commonly done in similar studies (Chizat et al., 2019).

# 3 Transfer Results in the Kernel Regime

In this section, we show how to generate adversarial examples from NTKs and discuss their similarity to the ones generated by the actual networks. Note that for network results, we restrict ourselves to wide networks initialized in the "lazy" regime with small learning rates (the "kernel regime").

## 3.1 Generation of Adversarial Examples for Infinitely Wide Neural Networks

Adversarial examples arise in the context of *classification*, while the NTK learning process is described by a regression as in Eq. (7). The arguably simplest way to align with the framework presented in Eq. (1) is to treat the outputs of the kernel similar to logits of a neural net, mapping them to a probability distribution via the sigmoid/softmax function and apply cross-entropy loss.

A simple calculation (see Appendix B, together with the generalization to the multi-class case) gives:

*The optimal one step adversarial example of a scalar, infinitely wide, neural network is given by*

$$\tilde{\mathbf{x}} = \mathbf{x} - y\epsilon \cdot \text{sign}\left(\nabla_{\mathbf{x}} f_t(\mathbf{x})\right), \tag{8}$$

for $\|\tilde{\mathbf{x}} - \mathbf{x}\|_\infty \leq \varepsilon$, where $\nabla_{\mathbf{x}} f_t(\mathbf{x}) = \nabla_{\mathbf{x}}\Theta(\mathbf{x}, \mathcal{X})^\top \Theta^{-1}(\mathcal{X}, \mathcal{X})(I - e^{-\lambda\Theta(\mathcal{X},\mathcal{X})t})\mathcal{Y}$.

One can conceive other ways to generate adversarial perturbations for the kernel, either by changing the loss function (as previously done in neural networks (e.g. (Carlini and Wagner, 2017))) or through a Taylor expansion around the test input, and we present such alternative derivations in Appendix B. However, in practice we observe little difference between that approach and the one presented here.

## 3.2 Transfer Results and Kernel Attacks

Predictions from NTK theory for infinitely wide neural networks have been used successfully for their large finite width counterparts, so it seems reasonable to conjecture that adversarial perturbations generated via the kernel as in Eq. (8) resemble those directly computed for the corresponding neural net as per Eq. (2). In particular, this would imply that adversarial perturbations derived from the NTK should not only fool the kernel machine itself, but also lead wide neural nets to misclassify.

While similar transfer results in different contexts have been observed indirectly, via the *effects* of the perturbation on metrics like accuracy (Yuan and Wu, 2021, Nguyen et al., 2021), we aim to look deeper to compare perturbations *directly*. High similarity would imply that *any* gradient based white-box attack on the neural net can be successfully mimicked by a "black-box" kernel derived attack.

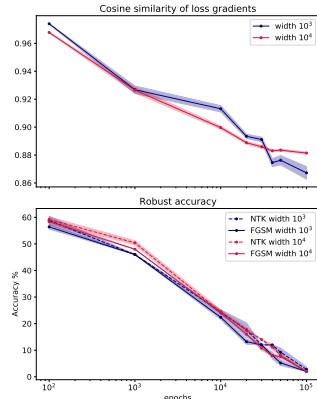

**Setting**. To this end, we train multiple two-layer neural networks on image classifications tasks extracted from MNIST and CIFAR-10 and compare adversarial examples generated by Eqs. (2) (attacking the neural network) and (8) (attacking the kernel). The networks are trained with small learning rate and are sufficiently large, so lie close to the NTK regime.

We track cosine similarity between the gradients of the loss from the NTK predictions and the gradients from the actual neural net as training evolves. Then, we generate adversarial perturbations from both the neural net and the kernel machine, and test whether those produced by the latter can fool the former. Full experimental details can be found in Appendix C.

Figure 3: **Top**. Cosine similarity between the loss gradient of the neural net and of the NTK prediction for the same time point. **Bottom**. Robust accuracy of neural net against its own adversarial examples (solid) and corresponding NTK examples (dashed). CIFAR10, car vs plane.

**Results**. Our experiments confirm a very strong alignment of loss gradients from the neural nets and the NTK across the whole duration of training, as can be seen in Fig. 3 (top). Then, as expected, kernel-generated attacks produce a similar drop in accuracy throughout training as the networks "own" white-box attacks, eventually driving robust accuracy to $0\%$, as seen in Fig. 3 (bottom). We reproduce these plots for MNIST in Appendix C, leading to similar conclusions.

When concerned with security aspects of neural nets, adversarial attacks are mainly characterised as either *white-box* or *black-box*

attacks (Papernot et al., 2017). White box attacks assume full access to the neural network and in particular its weights; prominent examples include FGSM/PGD attacks. Black box attacks, on the other hand, can only *query* the model to try to infer the loss gradient, either through training separate surrogate models (Papernot et al., 2016) or through carefully crafted input-output pairs fed to the target model (Chen et al., 2017, Ilyas et al., 2018, Andriushchenko et al., 2020). NTK theory and the experiments of this section suggest a threat model in which the attacker does not require access to the model or its weights, nor training of a substitute model. For fixed architecture and training data, all the information required for the computation of Eq. (8) is available at initialization, making the "NTK-attack" akin to a "training free" substitution attack, and, at least in the kernel-regime for wide nets considered here, as effective as white-box attacks.

## 4 NTK Eigenvectors Induce Robust and Non-Robust Features

This close connection between adversarial perturbations from the kernel and the corresponding neural net gives us the opportunity to bring to bear kernel tools on the study of adversarial robustness and its relation to features in a more direct fashion. Several recent works leverage properties of the NTK, and specifically its spectrum, to study aspects of approximation and generalization in neural networks (Arora et al., 2019a, Basri et al., 2019, Bietti and Mairal, 2019, Basri et al., 2020). Here we show how the spectrum relates to robustness and helps to clarify the notion of robust/non-robust features.

We define *features* induced by the eigendecomposition of the Gram matrix $\Theta(\mathcal{X}, \mathcal{X}) = \sum_{i=1}^{n} \lambda_i \mathbf{v}_i \mathbf{v}_i^\top$. We will be most interested in the *end* of training, when the model has access to all the features it can extract from the training data $\mathcal{X}$. As $t \to \infty$, Eq. (7) becomes $f_\infty(\mathbf{x}) = \Theta(\mathbf{x}, \mathcal{X})^\top \Theta(\mathcal{X}, \mathcal{X})^{-1} \mathcal{Y}$ and can be decomposed as $f_\infty(\mathbf{x}) = \Theta(\mathbf{x}, \mathcal{X})^\top \sum_{i=1}^{n} \lambda_i^{-1} \mathbf{v}_i \mathbf{v}_i^\top \mathcal{Y} = \sum_{i=1}^{n} f^{(i)}(\mathbf{x})$, where

$$f^{(i)} : \mathbb{R}^d \to \mathbb{R}^k, \ f^{(i)}(\mathbf{x}) := \lambda_i^{-1} \Theta(\mathbf{x}, \mathcal{X})^\top \mathbf{v}_i \mathbf{v}_i^\top \mathcal{Y}. \tag{9}$$

Each $f^{(i)}$ can be seen as a *unique feature* captured from the (training) data. Note that these functions map the input to the output space, thus matching the definitions of Sec. 2.2. Also observe that all $f^{(i)}$'s jointly recover the original prediction of the model, while each one, intuitively, should contribute something different to it.

Importantly, these features induce a decomposition of the gradient of the loss into parts, each representing gradients of a unique feature as already advertised in Fig. 1. The binary case is particularly elegant as it gives rise to a linear decomposition of the gradient as

$$\nabla_\mathbf{x} \mathcal{L}(f_\infty(\mathbf{x}), y) = \sum_{i=1}^{n} \alpha_i \nabla_\mathbf{x} \mathcal{L}(f^{(i)}(\mathbf{x}), y), \tag{10}$$

for some $\alpha_i$ depending on $\mathbf{x}$ and $y$ (see Appendix D). But if $f^{(i)}$'s are features, how do they look like?

**Feature properties of common architectures:** With these definitions in place, we can now analyze the characteristics of features for commonly used architectures, leveraging their associated NTK. To be consistent with the previous section, we consider classification problems from MNIST (10 classes) and CIFAR-10 (car vs airplane). We compose the Gram matrices from the whole training dataset (50000 and 10000, respectively), and compute the different feature functions $f^{(i)}$ using the eigendecomposition of the matrix. We estimate the **usefulness** of a feature $f^{(i)}$ by measuring its accuracy on a hold-out validation set, and its **robustness** by perturbing each input of this set, using an FGSM attack on feature $f^{(i)}$. We consider several different Fully Connected and Convolutional Kernels, whose expressions are available through the Neural Tangents library (Novak et al., 2020), built on top of JAX (Bradbury et al., 2018). We summarize our findings on how these features behave:



Figure 4: Non-robust, useful features earlier and later in the spectrum, for CIFAR10 car and plane.

*Functions $f^{(i)}$ represent visually distinct features.* We visualise each feature $f^{(i)}$ by plotting its gradient with respect to $\mathbf{x}$. Fig. 2 shows the gradient of the first 5 features for various architectures for a specific image from the CIFAR-10 dataset. We observe that features are fairly consistent across models, and they are interpretable: for example the 4th feature seems to represent the dominant color of an image, while the 5th one seems to be capturing horizontal edges.

*Networks use both robust and non-robust features for prediction.* It has been speculated that neural networks trained in a standard (non adversarial) fashion rely on both robust and non-robust features. Our feature definition in Eq. (9) shows that this is indeed the case. The NTK of common neural networks consists of both robust features that match human expectations, such as the ones depicted in Fig. 2, but also on features that are predictive of the true label, while not being robust to adversarial perturbations of the input (Fig. 4). Fig. 2 depicts the first 100 features of a fully connected and a convolutional tangent kernel in Usefulness-Robustness space. The upper left region of the plots shows a large amount of useful, yet non-robust features. These features seem random to human observers.

*Robustness lies at the top.* We observe in Fig. 2 that features corresponding to the top eigenvectors tend to be robust. This is consistent among different models and between the two datasets (see Appendix D). Since these eigenvectors are the ones fitted first during training (Arora et al., 2019a, Jacot et al., 2018), it is no wonder that the loss gradient evolves from coherence to noise, as observed in Fig. 1b (in the Appendix). This also explains the apparent trade-off between robustness and accuracy of neural networks as training progresses: useful, robust features are fitted first, followed by useful, but non-robust ones. This ties in well with both empirical findings (Rahaman et al., 2019) and theoretical case studies (Basri et al., 2019, Bietti and Mairal, 2019, Basri et al., 2020) that demonstrate that low frequency *functions* are fitted first during training and provide favorable generalization properties and we would associate robust features with these low-frequency parts (in function space).

*Robust features alone are not enough.* In light of these findings, it might be reasonable to conjecture that we could obtain robust models by retaining the robust features of the prediction, while discarding the non-robust ones. The spectral approach gives a principled way to disentangle features and create kernel machines keeping only the robust ones. Our results show that in general it is not possible to obtain non-trivial performance without compromising robustness in this fashion, strengthening the case for the necessity of data augmentation in the form of adversarial training (see Appendix D.3).

## 5 Kernel Dynamics during Adversarial Training

Given the apparent necessity for adversarial training to produce robust models, how does it achieve this goal? To shed some light on this fundamental question, we depart from the "lazy" NTK regime and study the evolution of the NTK of adversarially trained models. For a neural network trained with gradient descent, as the learning rate $\eta \to 0$, the continuous time dynamics can be written as

$$\frac{\partial w}{\partial t} = -\eta \nabla_w \mathcal{L} = -\eta \nabla_w f^\top \frac{\partial \mathcal{L}}{\partial f} \quad \text{and} \quad \frac{\partial f}{\partial t} = -\eta \underbrace{\nabla_w f \nabla_w f^\top}_{\Theta_t} \frac{\partial \mathcal{L}}{\partial f}. \tag{11}$$

In the NTK regime, this kernel $\Theta_t$ remains fixed at its initial value. However, outside this regime, it has been demonstrated, both empirically (Geiger et al., 2020, Fort et al., 2020, Baratin et al., 2021, Ortiz-Jiménez et al., 2021) and theoretically (Atanasov et al., 2022), that $\Theta_t$ is not constant during training, and is changing as the weights move. In adversarial training, moreover, there is the additional effect that at each weight update, the data changes as well. For that reason, understanding the dynamics of adversarial training requires tracking the evolution of a kernel $\Theta_t(\mathcal{X}_t, \mathcal{X}_t)$, where $\mathcal{X}_t$ denotes the current (mini) batch of training data. Notice that the tangent vector $\nabla_w f(\mathcal{X}_t)$ is still describing the instantaneous change of $f$ on the current batch of data, thus $\Theta_t(\mathcal{X}_t, \mathcal{X}_t)$ is informative of the local geometry of the function space, justifying its value as a quantity to be measured during adversarial training.

We train a deep convolutional architecture on CIFAR-10 (multiclass) with standard (SGD) and adversarial training using PGD with an $\ell_\infty$ constraint. Full implementations details and accuracy curves can be found in Appendix E, together with the reproduction of the same experiment on MNIST, where the observations are similar. We track the following quantities during training:

**Kernel distance.** We compare two kernels using a *scale invariant distance*, which quantifies the relative rotation between them, as used in other works studying NTK dynamics (e.g. Fort et al.

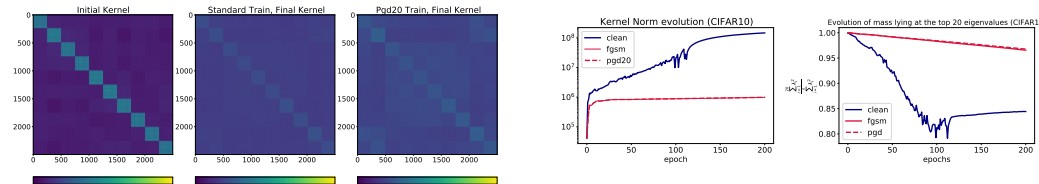

Figure 5: **Left**: Kernel Matrices for a mini batch of size 256. Left to Right: Kernel at initialization, Kernel after standard training, Kernel after adversarial training (20 PGD steps). The standard kernel grows significantly more than the adversarial one. **Right**: (a) Kernel Frobenius norm evolution, and (b) concentration on the top 20 eigenvalues during standard and adversarial training. Setting: CIFAR10, $\ell_\infty = 8/255$.

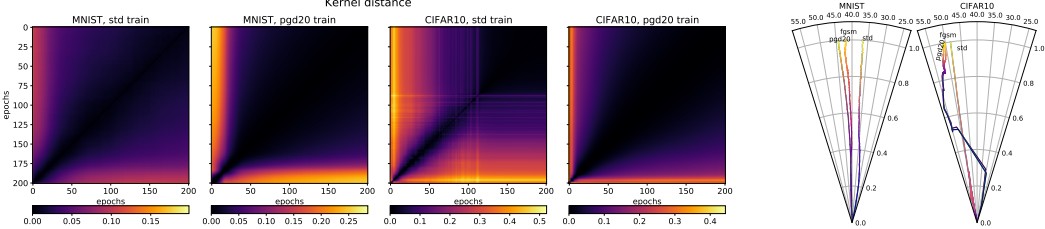

Figure 6: **Left:** Rotation (Eq. (12)) of the empirical NTK during standard, and adversarial training. Left to right: MNIST, standard, MNIST adversarial, CIFAR standard, CIFAR adversarial. **Right:** Kernel trajectories in polar space (Eq. (13)) for MNIST (left) and CIFAR10 (right). Darker colors indicate earlier epochs.

(2020)):

$$d(\Theta_i, \Theta_j) = 1 - \frac{\text{Tr}(\Theta_i \Theta_j^\top)}{\sqrt{\text{Tr}(\Theta_i \Theta_i^\top)}\sqrt{\text{Tr}(\Theta_j \Theta_j^\top)}}. \tag{12}$$

**Polar dynamics**. Zooming in on the change that the initial kernel undergoes, we define a *polar space* on which we measure the movement of the kernel:

$$r_t = \frac{\|\Theta_t - \Theta_0\|_F}{\|\Theta_f - \Theta_0\|_F}, \quad \theta_t = \arccos\left(1 - d(\Theta_t, \Theta_0)\right), \tag{13}$$

where $\Theta_0, \Theta_f$ are the initial and final kernel, respectively. Fig. 6 presents a heatmap of kernel distances at different time steps for both standard and adversarial training, as well as both training trajectories in polar space.

**Concentration on subspaces.** To quantify weight concentration on the top region of the spectrum, we track the (normalized) Frobenius norm of subspaces as $\sum_{i=1}^{p} \lambda_i^2 / \sum_{i=1}^{n} \lambda_i^2$, for various cut-offs $p$, where we have indexed the eigenvalues from largest to smallest. Fig. 5 depicts concentration on the top 20 eigenvalues during training.

Our findings show that similar to what has been reported in prior work (Fort et al., 2020), the kernel rotates significantly in the beginning of training and then slows down for both standard and adversarial training. However, in the latter case, this second phase begins a lot earlier. As Fig. 6 illuminates, the kernel moves a greater distance than when performing standard training, but after a few epochs it stops both rotating and expanding; note that this is not the case for standard training where the kernel increases its magnitude substantially later in training, and in fact grows to have a norm orders of magnitude larger than during adversarial training (see Fig. 5). In hindsight, this behavior is perhaps not surprising, as each element of the kernel measures similarity between data points, and a robust machine should be more conservative when estimating similarity. The observation that during adversarial training the kernel becomes relatively static relatively fast might indicate that *linear* dynamics govern the later phase of adversarial training. It has been observed in previous works (Geiger et al., 2020, Fort et al., 2020, Ortiz-Jiménez et al., 2021) that linearization after a few initial epochs of rapid rotation often closely matches performance of full network training. Our

results indicate that a similar phenomenon occurs even under the data shift of adversarial training (see Appendix E.1 for a study of linearized adversarial training), opening avenues to design robust machines more efficiently.

Moreover, endowed with the knowledge that at least for kernels trained with static data robust features lie at the top, we study polar dynamics of the top space only (see Fig. 8 in the Appendix) to observe that there is substantial rotation in this space, suggesting that robust features are learned early on not only during standard, but in particular during adversarial training. Even more interestingly, Fig. 5 demonstrates that not only the robust features change, but their relative weight as measured by the concentration on the top-20 space is increasing simultaneously relative to standard training as well, and remains large; in fact, significantly larger than during standard training. As each eigenvalue weights the importance of the corresponding feature on the final prediction, this implies that the kernel "learns" to depend more on the most robust features.

Put together, these findings reveal different kernel dynamics during standard and adversarial training: the kernel rotates much faster, expands much less and becomes "lazy" much earlier than during standard training. Fully understanding the properties of converged adversarial kernels remains an important avenue for future work, that might allow to design faster algorithms for robust classification.

## 6 Final Remarks

We have studied adversarial robustness through the lens of the NTK across multiple architectures and data sets both in the idealized NTK regime and the "rich feature" regime. When connecting the spectrum of the kernel with fundamental properties characterizing robustness our phenomenological study reveals a universal picture of the emergence of robust and non-robust features and their role during training. There are certain limitations and unexplored themes in our work; Sec. 3 argues that transferable attacks from the NTK may be as effective as white-box attacks, but this warrants an in-depth study across architectures, kernels and data sets (which has not been the main focus of this work). Sec. 4 visualises features for fairly simple models, since the computation of kernel derivatives is a costly procedure. It would be interesting to use our framework to visualise features from more complicated architectures. Finally, our work in Sec. 5 invites more research on the kernel at the end of adversarial training, similar to what has been done for standard models (Long, 2021).

We hope that our viewpoint can motivate further theoretical understanding of adversarial phenomena (such as transferability) and the design of better and/or faster adversarial learning algorithms, by further analyzing the kernels from robust deep neural networks.

## Acknowledgements

The authors would like to thank Jingtong Su, Alberto Bietti, Yunzhen Feng, and Artem Vysogorets for fruitful discussions and feedback in various stages of this work. NT thanks Dimitris Tsipras for a helpful discussion in the beginning of this project. The authors would like to acknowledge support through the National Science Foundation under NSF Award 1922658. This work was supported in part through the NYU IT High Performance Computing resources, services, and staff expertise.

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
