# What Can the Neural Tangent Kernel Tell Us About Adversarial Robustness? - Supplementary material

**Nikolaos Tsilivis**
Center for Data Science
New York University
nt2231@nyu.edu

**Julia Kempe**
Center for Data Science
New York University
kempe@nyu.edu

## A  Robust and Non-Robust features

The idea that data features are to be blamed for the adversarial weakness of machine learning models was proposed in (Ilyas et al., 2019, Tsipras et al., 2019). In particular, Ilyas et al. (2019) show that training with adversarially perturbed images labeled with the "wrong" label yields classifiers with non-trivial test performance ("learning from non-robust features only"), while, in a dual experiment, they demonstrate that standard training with "robustified" data (data that presumably are "denoised" from non-robust features) produces a classifier with non-trivial *robust* accuracy ("relies only on robust features"). Motivated by these observations, the authors propose a model of robust/non-robust features that are hidden in the data, and whose presence determines the eventual robustness of models. To accompany the definitions of Sec. 2.2, we extend them for multiclass classification, since Sec. 4 introduces our NTK feature framework for both binary and multiclass problems.

Let $\mathcal{D}$ be the data generating distribution, with $x \in \mathcal{X}$ (input space) and $y \in \{1, \ldots, k\}$ (action space). We define features $\phi : \mathcal{X} \to \mathbb{R}^k$ as functions from the input to the action space, and categorize them as follows, according to their performance as classifiers. Fix $\rho, \gamma \geq 0$:

1. $\rho$-**Useful** feature: A feature $\phi$ is called $\rho$-*useful* if

$$\mathbb{E}_{x,y\sim\mathcal{D}}\left[\mathbb{1}_{\{\arg\max_{i\in[k]}\phi_i(x)=y\}}\right] = \rho \tag{1}$$

2. $\gamma$-**Robust** feature: A feature $\phi$ is called $\gamma$-*robust* if it is predictive of the true label under any perturbation inside a bounded "ball" $\mathcal{B}$, that is if

$$\mathbb{E}_{x,y\sim\mathcal{D}}\left[\inf_{\delta\in\mathcal{B}}\mathbb{1}_{\{\arg\max_{i\in[k]}\phi_i(x+\delta)=y\}}\right] = \gamma \tag{2}$$

3. **Useful, non-robust** feature: A feature is called **useful, non-robust** if it confers an advantage above guessing the most likely label, i.e. $\exists \rho > \max_{i\in[k]} \mathbb{E}_{x,y\sim\mathcal{D}}[\mathbb{1}_{\{i=y\}}]$, but is $\gamma$-robust only for $\gamma \approx 0$ (within some precision).

The above framework was introduced by (Ilyas et al., 2019, Tsipras et al., 2019), and we have slightly adapted it in terms of accuracy as classifiers derived from features. Goh (2019) showed how such feature functions arise in a simple linear model, and proposed two mechanisms to construct useful, non-robust features. In (Allen-Zhu and Li, 2022), the authors view the weights of neural networks as features, and show that adversarial training "purifies/robustifies" them.

## B  Derivation of Adversarial Perturbations for Kernel Regression

In this section, we derive expressions for adversarial attacks on Neural Tangent Kernels presented in the main paper, as well as additional derivations obtained from first-order expansions around the input.

36th Conference on Neural Information Processing Systems (NeurIPS 2022).

## B.1 Adversarial Perturbations from Cross-Entropy Loss

We first derive the expression in Eq. (8) of the paper. Let $\mathbf{x} \in \mathbb{R}^d$ be an input to the NTK prediction

$$f_t(\mathbf{x}) = \Theta(\mathbf{x}, \mathcal{X})^\top \Theta^{-1}(\mathcal{X}, \mathcal{X})(I - e^{-\lambda\Theta(\mathcal{X}, \mathcal{X})t})\mathcal{Y}, \tag{3}$$

where $(\mathcal{X}, \mathcal{Y})$ is a dataset of size $n$. We consider the binary and the multiclass case separately.

In the **binary** case, where $y \in \{\pm 1\}$, we feed expression Eq. (3) to a sigmoid $\sigma(x) = (1 + e^{-x})^{-1}$ and maximize the cross entropy loss between the output and the true label:

$$\mathcal{L}(\mathbf{x}, y) = -\hat{y} \log\left(\sigma(f_t(\mathbf{x}))\right) - (1 - \hat{y}) \log\left(1 - \sigma(f_t(\mathbf{x}))\right), \tag{4}$$

where we set $\hat{y} = \frac{y+1}{2}$ to lie in $\{0, 1\}$. We compute the gradient of the loss with respect to $\mathbf{x}$:

$$
\begin{aligned}
\nabla_{\mathbf{x}}\mathcal{L}(\sigma(f_t(\mathbf{x})), y) &= -\frac{\hat{y}}{\sigma(f_t(\mathbf{x}))}\nabla_{\mathbf{x}}\sigma(f_t(\mathbf{x})) + \frac{(1 - \hat{y})}{1 - \sigma(f_t(\mathbf{x}))}\nabla_{\mathbf{x}}\sigma(f_t(\mathbf{x})) \\
&= \frac{\sigma(f_t(\mathbf{x})) - \hat{y}}{\sigma(f_t(\mathbf{x}))(1 - \sigma(f_t(\mathbf{x})))}\nabla_{\mathbf{x}}\sigma(f_t(\mathbf{x})) = (\sigma(f_t(\mathbf{x})) - \hat{y})\nabla_{\mathbf{x}}f_t(\mathbf{x}).
\end{aligned} \tag{5}
$$

So the optimal one-step attack, under an $\ell_\infty$ adversary, reduces to computing perturbation

$$\boldsymbol{\eta} = \epsilon \cdot \text{sign}\left((\sigma(f_t(\mathbf{x})) - \hat{y})\nabla_{\mathbf{x}}f_t(\mathbf{x})\right) = -\epsilon y \cdot \text{sign}\left(\nabla_{\mathbf{x}}f_t(\mathbf{x})\right), \tag{6}$$

since $\sigma(u) \in (0, 1)$ for all $u \in \mathbb{R}$.

In the case of a **k-class** classification problem with one hot labels $\mathcal{Y} \in \mathbb{R}^{n \times k}$, we can express the cross entropy loss between the NTK predictions Eq. (3) and the labels as:

$$\mathcal{L}(\mathbf{x}, y) = -\log \frac{e^{f_{t,y}(\mathbf{x})}}{\sum_{r=1}^{k} e^{f_{t,r}(\mathbf{x})}} = -f_{t,y}(\mathbf{x}) + \log \sum_{r=1}^{k} e^{f_{t,r}(\mathbf{x})}, \tag{7}$$

where $f_{t,r}$ denotes the $r$-th output of Eq. (3). Computing the loss gradient as before yields the optimal perturbation $\boldsymbol{\eta}$,

$$\boldsymbol{\eta} = \epsilon \cdot \text{sign}\left(-\nabla_{\mathbf{x}}f_{t,y}(\mathbf{x}) + \frac{\sum_{r=1}^{k} e^{f_{t,r}(\mathbf{x})}\nabla_{\mathbf{x}}f_{r,y}(\mathbf{x})}{\sum_{r=1}^{k} e^{f_{r,t}(x)}}\right). \tag{8}$$

The above calculations allow us to speed up the computation of the attacks in the case of NTKs with closed form expression, since the gradient

$$\nabla_{\mathbf{x}}f_{t,r}(\mathbf{x}) = D\Theta(\mathbf{x}, \mathcal{X})^\top \Theta^{-1}(\mathcal{X}, \mathcal{X})(I - e^{-\lambda\Theta(\mathcal{X}, \mathcal{X})t})\mathcal{Y}_{:,r}, \tag{9}$$

with D being the Jacobian of $\Theta$ wrt to $\mathbf{x}$, can be pre-computed, without the need for auto-differentiation tools. We leverage this in the experiments of Sec. 3.

## B.2 Alternative Approaches to Generate Perturbations

One can derive other perturbation variants by changing the loss function from cross-entropy to other functions studied in the literature in this context (e.g. (Carlini and Wagner, 2017)). Alternatively, we can study the output $f_t(x)$ on a test input $x$ directly to devise strategies to most efficiently perturb it, using a Taylor expansion around the input, leading to a linear expression (shown here for scalar kernels):

$$f(\mathbf{x} + \boldsymbol{\eta}) \approx f(\mathbf{x}) + \boldsymbol{\eta}^T \mathbf{z}, \tag{10}$$

for some $z \in \mathbb{R}^d$ that depends on the training data and the NTK kernel only.

**Binary case:** Suppose we would like to evaluate a model described by Eq. (7) at the end of training,

$$f_\infty(\mathbf{x}) = \Theta(\mathbf{x}, \mathcal{X})^\top \Theta(\mathcal{X}, \mathcal{X})^{-1}\mathcal{Y} \tag{11}$$

on *slightly* perturbed variations of the original *training* data. Then, slightly abusing notation, we set, $\tilde{\mathcal{X}} = \mathcal{X} + \boldsymbol{\epsilon}$, that is $\tilde{\mathbf{x}}_i = \mathbf{x}_i + \boldsymbol{\eta}_i$ for all $\mathbf{x}_i \in \mathcal{X}$ for small, but unknown, perturbations $\boldsymbol{\eta}_i$. By

taking a first-order Taylor expansion in the perturbation, we can write the $ij$-th element of $\Theta(\tilde{\mathcal{X}}, \mathcal{X})$ as follows:

$$\Theta(\tilde{\mathbf{x}}_i, \mathbf{x}_j) = \Theta(\mathbf{x}_i + \boldsymbol{\eta}_i, \mathbf{x}_j) \approx \Theta(\mathbf{x}_i, \mathbf{x}_j) + \nabla_{\mathbf{x}_i}\Theta(\mathbf{x}_i, \mathbf{x}_j)^T \boldsymbol{\eta}_i. \tag{12}$$

For each row $\underbrace{\Theta_{i,:}(\tilde{\mathcal{X}}, \mathcal{X})}_{\in \mathbb{R}^{1 \times n}}$ we obtain:

$$\Theta_{i,:}(\tilde{\mathcal{X}}, \mathcal{X})^T = \Theta_{i,:}(\mathcal{X}, \mathcal{X})^T + \underbrace{\begin{pmatrix} \nabla_{\mathbf{x}_i}\Theta(\mathbf{x}_i, \mathbf{x}_1)^T \\ \nabla_{\mathbf{x}_i}\Theta(\mathbf{x}_i, \mathbf{x}_2)^T \\ \vdots \\ \nabla_{\mathbf{x}_i}\Theta(\mathbf{x}_i, \mathbf{x}_n)^T \end{pmatrix}}_{\mathbf{A}_i \in \mathbb{R}^{n \times d}} \boldsymbol{\eta}_i. \tag{13}$$

Hence, $\Theta(\tilde{\mathcal{X}}, \mathcal{X})$ can be written as $\Theta(\mathcal{X}, \mathcal{X}) + \boldsymbol{\Delta}$ for a perturbation matrix $\boldsymbol{\Delta}$, with $i$-th row $\boldsymbol{\Delta}_{i,:} = \boldsymbol{\eta}_i^T \mathbf{A}_i^T$. Substituting into Eq. (11), we get:

$$f(\tilde{\mathcal{X}}) = (\Theta(\mathcal{X}, \mathcal{X}) + \boldsymbol{\Delta})\Theta(\mathcal{X}, \mathcal{X})^{-1}\mathcal{Y} = \mathcal{Y} + \boldsymbol{\Delta}\Theta(\mathcal{X}, \mathcal{X})^{-1}\mathcal{Y}. \tag{14}$$

Thus, the output of the model on $\tilde{\mathbf{x}}_i$ is:

$$\begin{aligned} f(\tilde{\mathbf{x}}_i) &= y_i + \boldsymbol{\Delta}_i \Theta(\mathcal{X}, \mathcal{X})^{-1}\mathcal{Y} \\ &= y_i + \boldsymbol{\eta}_i^T \mathbf{A}_i^T \Theta(\mathcal{X}, \mathcal{X})^{-1}\mathcal{Y} =: y_i + \boldsymbol{\eta}_i^T \mathbf{z}_i, \end{aligned} \tag{15}$$

leading to the linear expression advertised in Eq. (10). The adversarial perturbation $\boldsymbol{\eta}_i$ changes the output by $\boldsymbol{\eta}_i^T \mathbf{z_i} = \boldsymbol{\eta}_i^T \mathbf{A}_i^T \Theta(\mathcal{X}, \mathcal{X})^{-1}\mathcal{Y}$, an expression which allows us to *compute* the adversarial perturbation to maximally change the output within the desired constraints on $\boldsymbol{\eta_i}$.

Since Eq. (11) describes regression models with LSE ($L_2$-loss), while adversarial examples typically are studied for classification models, we use thresholding (i.e. taking the sign of the output in the case of binary $\{-1, 1\}$ classification tasks) or by outputting the maximum prediction (in the case of multiclass problems) to turn Eq. (11) into a classifier.

Inspecting Eq. (15), maximal "confusion" of the classification model is achieved by aligning $\boldsymbol{\eta}_i$ with $-y_i \mathbf{z_i}$ (directed towards the decision boundary). In case of the commonly used $\ell_\infty$ restriction, i.e. $\|\boldsymbol{\eta}_i\|_\infty \leq \epsilon$, the optimal adversarial perturbation is given by:

$$\boldsymbol{\eta}_i = -\epsilon y_i \cdot \text{sign}(\mathbf{A}_i^T \Theta(\mathcal{X}, \mathcal{X})^{-1}\mathcal{Y}). \tag{16}$$

The computation of this optimal adversarial perturbation requires an expression for the NTK and its gradient with respect to the training data. For models where an *analytical* expression of the NTK is available, only access to the labeled training data is necessary (as presented, for instance, in Sec. C). In more complicated models or those that deviate from the assumptions for Eq. (11) one can compute an *empirical* kernel by sampling over kernels at initialization over a few instances and obtain the matrices $\mathbf{A_i}$ with autodifferentiation tools.

Eq. (15) has been derived for perturbations of the *training* data. Consider now the case when we evaluate Eq. (11) on perturbations of unseen *test* data, that is on $\tilde{\mathcal{X}} + \boldsymbol{\epsilon}$. Then, Eq. (14) becomes:

$$f(\tilde{\mathcal{X}} + \boldsymbol{\epsilon}) = (\Theta(\tilde{\mathcal{X}}, \mathcal{X}) + \boldsymbol{\Delta})\Theta(\mathcal{X}, \mathcal{X})^{-1}\mathcal{Y} = f(\tilde{\mathcal{X}}) + \boldsymbol{\Delta}\Theta(\mathcal{X}, \mathcal{X})^{-1}\mathcal{Y}. \tag{17}$$

Again, solely the second term depends on the perturbation, so we proceed by choosing a maximally perturbing direction as before. The only difference lies in the matrix $\boldsymbol{\Delta}$ that now depends on the test set $\tilde{\mathcal{X}}$

$$\boldsymbol{\Delta}_{i,:} = \boldsymbol{\eta}_i^T \begin{pmatrix} \nabla_{\tilde{\mathbf{x}}_i}\Theta(\tilde{\mathbf{x}}_i, \mathbf{x}_1)^T \\ \nabla_{\tilde{\mathbf{x}}_i}\Theta(\tilde{\mathbf{x}}_i, \mathbf{x}_2)^T \\ \vdots \\ \nabla_{\tilde{\mathbf{x}}_i}\Theta(\tilde{\mathbf{x}}_i, \mathbf{x}_n)^T \end{pmatrix}^T \tag{18}$$

In practice, an adversary can calculate the NTK $\Theta(\mathcal{X}, \mathcal{X})$ offline and calculate the optimal perturbation on a new test input $\tilde{\mathbf{x}}_i$ by computing the corresponding row of the matrix $\boldsymbol{\Delta}$. Importantly, no information on the test data *labels* is needed.

**Multiclass case:** We adapt the derivations of the binary case to the setting where the output dimension is larger than one in the underlying regression setting (see below), resulting in a multiclass classifier. This leads to the multi-dimensional analogue of the linear Eq. (10) for $f(x) \in \mathbb{R}^k$, $y \in \mathbb{R}^k$:

$$f(\mathbf{x}_i + \boldsymbol{\eta}_i) = \mathbf{y}_i + \begin{pmatrix} \boldsymbol{\eta}_i^T \mathbf{z}_1 \\ \boldsymbol{\eta}_i^T \mathbf{z}_2 \\ \vdots \\ \boldsymbol{\eta}_i^T \mathbf{z}_k \end{pmatrix}. \tag{19}$$

Again, the $z \in \mathbb{R}^d$ can be computed from the NTK and its derivative as well as the training data labels. Exactly analogous considerations as in the binary case allow to adapt this expression to perturbations of the *test* data.

At this point we have a choice of how to adversarially perturb the classifier to achieve the largest effect on the network output. We present the two most obvious methods.

*Max-of-$\ell_1$ perturbation:* Similar in spirit to traditional approaches in adversarial attacks (Carlini and Wagner (2017)) we choose $\boldsymbol{\eta}_i$ such as to most efficiently decrease the correct response $r^* = \arg\max_j (\mathbf{y}_i)_j$ while maximally increasing one of the false responses $r \neq r^*$. The solution is given by:

$$\boldsymbol{\eta}_i = \varepsilon \cdot \text{sign}(\arg \max_{r=1, r \neq r^\star}^{k} \|\mathbf{z}_r - \mathbf{z}_{r^\star}\|_1). \tag{20}$$

It is obtained by solving

$$\boldsymbol{\eta}_i = \arg \max_{\|\boldsymbol{\eta}_i\|_\infty \leq \epsilon} \max_{r=1, r \neq r^\star}^{k} f_r(\mathbf{x}_i + \boldsymbol{\eta}_i) - f_{r^\star}(\mathbf{x}_i + \boldsymbol{\eta}_i).$$

Then

$$\begin{aligned} \boldsymbol{\eta}_i &= \arg \max_{\|\boldsymbol{\eta}_i\|_\infty \leq \epsilon} \max_{r=1, r \neq r^\star}^{k} \boldsymbol{\eta}_i^T \mathbf{z}_r - \boldsymbol{\eta}_i^T \mathbf{z}_{r^\star} \\ &= \arg \max_{\|\boldsymbol{\eta}_i\|_\infty \leq \epsilon} \max_{r=1, r \neq r^\star}^{k} \boldsymbol{\eta}_i^T (\mathbf{z}_r - \mathbf{z}_{r^\star}) \\ &= \varepsilon \cdot \text{sign}(\arg \max_{r=1, r \neq r^\star}^{k} \|\mathbf{z}_r - \mathbf{z}_{r^\star}\|_1). \end{aligned} \tag{21}$$

*Sum-of-$\Delta z$ perturbation:* For one-hot vectors $\mathbf{y}_i$ we could, instead, maximize the cross-entropy between the labels and the new outputs, thus choosing to produce a maximally mixed output. If $r^*$ is the correct label, this yields

$$\boldsymbol{\eta}_i = \varepsilon \cdot \text{sign}(\sum_{r \neq r^*}^{n} (\mathbf{z}_r - \mathbf{z}_{r^\star})). \tag{22}$$

derived as follows

$$\begin{aligned} L_{ce}(f(\mathbf{x}_i + \boldsymbol{\eta}), \mathbf{y}_i) &= -\sum_{r=1}^{k} y_i^{(r)} \log \left( \frac{e^{y_i^{(r)} + \boldsymbol{\eta}_i^T \mathbf{z}_r}}{\sum_{r'=1}^{k} e^{y_i^{(r')} + \boldsymbol{\eta}_i^T \mathbf{z}_r'}} \right) \\ &= -\log \left( \frac{e^{1 + \boldsymbol{\eta}_i^T \mathbf{z}_{r^\star}}}{\sum_{r=1, r \neq r^\star}^{k} e^{\boldsymbol{\eta}_i^T \mathbf{z}_r} + e^{1 + \boldsymbol{\eta}_i^T \mathbf{z}_{r^\star}}} \right) \\ &= \log \left( \sum_{r \neq r^\star} e^{\boldsymbol{\eta}_i^T (\mathbf{z}_r - \mathbf{z}_{r^\star}) - 1} + 1 \right). \end{aligned} \tag{23}$$

Maximizing this cross entropy amounts to maximizing

$$\sum_{r \neq r^\star} e^{\boldsymbol{\eta}_i^T (\mathbf{z}_r - \mathbf{z}_{r^\star})}.$$

For small perturbations we can develop the exponential to first order[1], which leads to finding the maximum of

$$\boldsymbol{\eta}_i^T \sum_{r \neq r^\star} (\mathbf{z}_r - \mathbf{z}_{r^\star}),$$

yielding Eq. (22).

*Derivation of Eq.* (19)*:* While we remain with $\mathcal{X} \in \mathbb{R}^{n \times d}$ as in the binary case, the other quantities change as $\mathcal{Y} \in \mathbb{R}^{nk}$, $f(\mathcal{X}) \in \mathbb{R}^{nk}$ and $\Theta(\mathcal{X}, \mathcal{X}) \in \mathbb{R}^{nk \times nk}$, i.e. for each data pair $(\mathbf{x}_i, \mathbf{x}_j)$ we have $\Theta(\mathbf{x}_i, \mathbf{x}_j) \in \mathbb{R}^{k \times k}$. Let $\Theta_{lm}(\mathbf{x}_i, \mathbf{x}_j)$ denote the entry of $\Theta(\mathbf{x}_i, \mathbf{x}_j)$ that corresponds to the $l$-th and the $m$-th output of the model (evaluated at $\mathbf{x}_i$ and $\mathbf{x}_j$). Then, with similar reasoning that led to Eq. (12) we now obtain:

$$\Theta_{lm}(\mathbf{x}_i + \boldsymbol{\eta}, \mathbf{x}_j) \approx \Theta_{lm}(\mathbf{x}_i, \mathbf{x}_j) + \nabla_{\mathbf{x}_i}\Theta_{lm}^T(\mathbf{x}_i, \mathbf{x}_j)\boldsymbol{\eta}. \tag{24}$$

For the prediction of the model on the whole dataset, we have:

$$f(\tilde{\mathbf{X}}) = \mathcal{Y} + \underbrace{\boldsymbol{\Delta}}_{\in \mathbb{R}^{nk \times nk}} (\Theta(\mathbf{X}, \mathbf{X}))^{-1}\mathcal{Y}, \tag{25}$$

which for a given sample $\mathbf{x}_i$ gives:

$$f(\mathbf{x}_i + \boldsymbol{\eta}_i) = \underbrace{\mathbf{y}_i}_{\in \mathbb{R}^k} + \boldsymbol{\Delta}_{ik:(i+1)k,:} (\Theta(\mathbf{X}, \mathbf{X}))^{-1}\mathcal{Y}, \tag{26}$$

where $\boldsymbol{\Delta}_{(i-1)k:ik,:}$ is equal to

$$\underbrace{\begin{pmatrix} \overbrace{(\nabla_{\mathbf{x}_i}\Theta_{11}(\mathbf{x}_i, \mathbf{x}_1) \quad \dots \quad \nabla_{\mathbf{x}_i}\Theta_{1k}(\mathbf{x}_i, \mathbf{x}_1) \quad \nabla_{\mathbf{x}_i}\Theta_{11}(\mathbf{x}_i, \mathbf{x}_2) \quad \dots \quad \nabla_{\mathbf{x}_i}\Theta_{1k}(\mathbf{x}_i, \mathbf{x}_n))^T}^{\in \mathbb{R}^{nk \times d}}\boldsymbol{\eta}_i \\ (\nabla_{\mathbf{x}_i}\Theta_{21}(\mathbf{x}_i, \mathbf{x}_1) \quad \dots \quad \nabla_{\mathbf{x}_i}\Theta_{2k}(\mathbf{x}_i, \mathbf{x}_1) \quad \nabla_{\mathbf{x}_i}\Theta_{21}(\mathbf{x}_i, \mathbf{x}_2) \quad \dots \quad \nabla_{\mathbf{x}_i}\Theta_{2k}(\mathbf{x}_i, \mathbf{x}_n))^T\boldsymbol{\eta}_i \\ \vdots \\ (\nabla_{\mathbf{x}_i}\Theta_{k1}(\mathbf{x}_i, \mathbf{x}_1) \quad \dots \quad \nabla_{\mathbf{x}_i}\Theta_{kk}(\mathbf{x}_i, \mathbf{x}_1) \quad \nabla_{\mathbf{x}_i}\Theta_{k1}(\mathbf{x}_i, \mathbf{x}_2) \quad \dots \quad \nabla_{\mathbf{x}_i}\Theta_{kk}(\mathbf{x}_i, \mathbf{x}_n))^T\boldsymbol{\eta}_i \end{pmatrix}}_{\in \mathbb{R}^{k \times nk}}.$$

$$\tag{27}$$

## C  Transfer Results for Wide Two-Layer Networks

In this section, we present additional experimental details for Sec. 3.2 and show the results of the experiments on MNIST. We train two-layer neural networks of the form

$$f(\mathbf{x}) = \frac{1}{\sqrt{m}}\mathbf{A}\max(\mathbf{W}\mathbf{x}, 0), \quad \mathbf{A} \in \{\pm 1\}^{m \times k}, \mathbf{W} \sim \mathcal{N}(0, 0.01^2 I_{m \times d}), \tag{28}$$

where the first layer is initialized with the normal distribution, the second layer is frozen to its initial random values in $\{\pm 1\}$, and $m$ denotes the width of the network. The NTK of this architecture is given by

$$\Theta(\mathbf{x}_i, \mathbf{x}_j) = \left(\frac{1}{2} - \frac{\arccos\left(\frac{\mathbf{x}_i^\top \mathbf{x}_j}{\|\mathbf{x}_i\|\|\mathbf{x}_j\|}\right)}{2\pi}\right)\mathbf{x}_i^\top \mathbf{x}_j. \tag{29}$$

We choose this family of models in order to be consistent with early works that analyzed training and generalization properties of neural networks in the NTK regime (Arora et al., 2019, Du et al., 2019). We perform experiments on image classification on MNIST and on a binary task extracted from CIFAR-10 (car vs airplane). We train the networks in a regression fashion, minimizing the $\ell_2$ loss between the predictions and one-hot vectors, using full-batch gradient descent on the entire dataset (full training data for MNIST and 5K images for each of car and airplane in binary CIFAR). We keep the learning rate fixed to $10^{-2}$ and vary the width of the network in $\{10^3, 10^4\}$. We train 3 networks for each dataset until convergence ($10^5$ epochs), each initialized with a different random

---

[1]The resulting expression for the maximum also holds when developing to second order.

seed. When we measure quantities from the neural net, we subtract the initial prediction $f_0$, since the NTK expression Eq. (3) does not take the initialization of the network into account. When attacking the models ($\ell_\infty$ attacks), we use perturbation budget $\epsilon = 0.3$ for MNIST and $\epsilon = 8/255$ for CIFAR-10. The experiments are performed with PyTorch (Paszke et al., 2019).

For each model, we calculate the loss gradients with respect to the input during training, and compare them to those derived for the NTK in Eqs. (5) and (8) for the binary and the multiclass task, respectively, using cosine similarity:

$$\frac{\nabla_x \mathcal{L}(f_t, y)^\top \nabla_x \mathcal{L}(f - f_0, y)}{\|\nabla_x \mathcal{L}(f_t, y)\|_2 \|\nabla_x \mathcal{L}(f - f_0, y)\|_2},\tag{30}$$

where $f_t$ is the NTK prediction defined in Eq. (3), $f$ denotes the output of the neural net and $f_0$ is the initial prediction of the neural net (prior to training). In order to match the time-scales, we manually align the networks on epoch $= 10^3$ with a time-point for the NTK, and based on this number, we match the rest of the epochs assuming linear dependence (as theory predicts - Eq. (3)). Fig. 1a shows cosine similarity of loss gradients and robust accuracy of the network (evaluated against its own adversarial examples, and those from the NTK) for MNIST. Fig. 1b illustrates the similarity of loss gradients of neural nets and their NTKs for 3 different epochs.

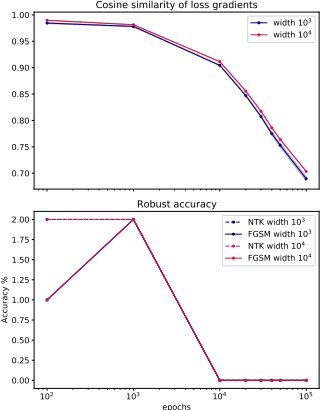

(a) Comparison of NTK and neural net derived quantities for digit recognition (MNIST) during training.

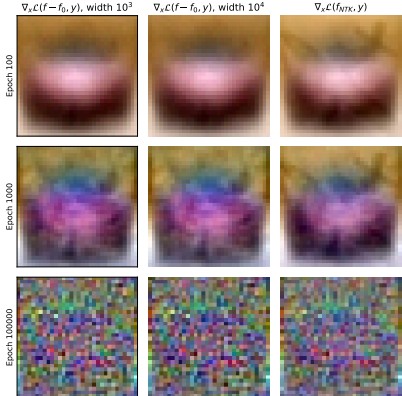

(b) Loss gradients.

Figure 1: Visualizing the similarity of NTK and NN adversarial quantities. (a) **Top**. Cosine similarity between the loss gradient of the neural net and of the NTK prediction for the same time point (MNIST). **Bottom**. Robust accuracy of neural net against its own adversarial examples (solid) and corresponding NTK examples (dashed) for MNIST. Blue and red lines overlap in the second plot, and the effect of the random seed is insignificant. (b) Illustration of the similarity of loss gradients for NTK (**right** column) and neural nets of width $10^3$ (**left**) and $10^4$(**middle**) for a specific image extracted from CIFAR-10. Columns show gradients for different epochs ($10^2, 10^3, 10^4$, respectively).

Notice the very small discrepancy between the loss gradients of different networks (initialized with different random seeds) in Fig. 1a. They are all centered around the loss gradient of the NTK, a manifestation of transferability of adversarial examples, at least for models with the same architecture. The NTK framework might possibly provide a wider explanation of this phenomenon, also across architectures. For instance, for fully connected kernels, the NTK expression for kernels of depth $l$ is a relatively simple function of expressions for depth $l - 1$ (Jacot et al., 2018, Bietti and Mairal, 2019) which could explain transferability across architectures of varying depth.

## D    NTK Features: Additional Details

In this section, we present additional material for Sec. 4; we show derivations that are missing from the main text, and complement the plots by showing the same information for more architectures and datasets.

### D.1 Loss Gradient Decomposition

First, recall our definitions of features from Sec. 4. Let $\mathcal{X}, \mathcal{Y}$ be a dataset, where $\mathcal{X} \in \mathbb{R}^{n \times d}$ and $\mathcal{Y} \in \{\pm 1\}^n$ (binary classification). Then, kernel regression on this dataset gives predictions of the form $f_\infty(\mathbf{x}) = \Theta(\mathbf{x}, \mathcal{X})^\top \Theta(\mathcal{X}, \mathcal{X})^{-1} \mathcal{Y}$. Given, the eigendecomposition of the Gram Matrix $\Theta(\mathcal{X}, \mathcal{X})$, we can decompose the prediction as follows

$$f_\infty(\mathbf{x}) = \Theta(\mathbf{x}, \mathcal{X})^\top \sum_{i=1}^{n} \lambda_i^{-1} \mathbf{v}_i \mathbf{v}_i^\top \mathcal{Y} = \sum_{i=1}^{n} f^{(i)}(\mathbf{x}), \tag{31}$$

where $f^{(i)} : \mathbb{R}^d \to \mathbb{R}^k, f^{(i)}(\mathbf{x}) = \lambda_i^{-1} \Theta(\mathbf{x}, \mathcal{X})^\top \mathbf{v}_i \mathbf{v}_i^\top \mathcal{Y}$. Notably, this means that the gradient of the cross entropy loss can be also understood as a composition of gradients coming from these features, as the following proposition shows.

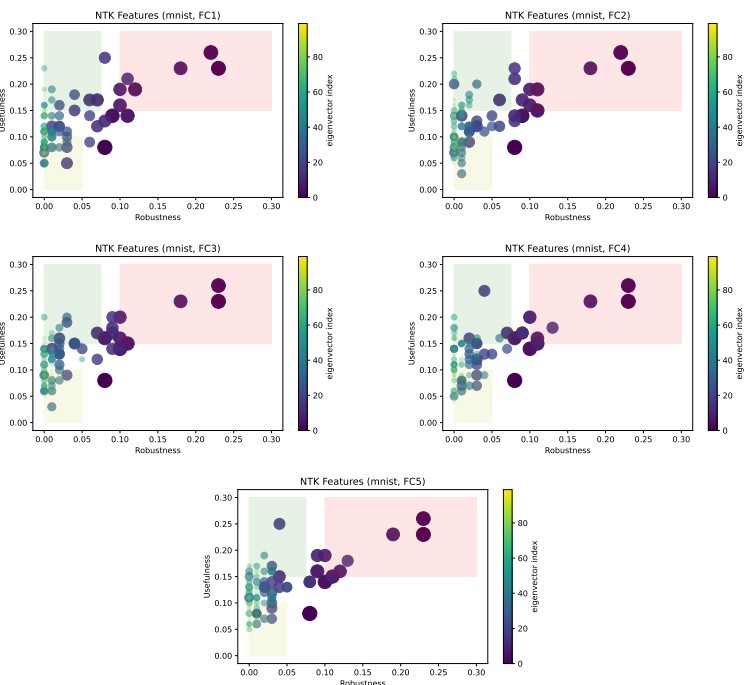

Figure 2: Robustness Usefulness space for various kernels, MNIST multiclass. The axes lie in $[0, 1]$. "Useful" features have usefulness above 0.1 (the random guessing probability for our balanced data set). The colored red, green and yellow boxes are arbitrary, meant to visually distinguish useful-robust from other features.

**Proposition 1.** *The loss gradient of $f_\infty$ can be decomposed as follows:*

$$\nabla_{\mathbf{x}} \mathcal{L}(f_\infty(\mathbf{x}), y) = \sum_{i=1}^{n} \alpha_i \nabla_{\mathbf{x}} \mathcal{L}(f^{(i)}(\mathbf{x}), y), \tag{32}$$

*where $\alpha_i$ is a quantity that depends on $\mathbf{x}, y$.*

*Proof.* Recall from Eq. (5), that $\nabla_{\mathbf{x}} \mathcal{L}(f(\mathbf{x}), y) = \left( \sigma(f(x)) - \frac{y+1}{2} \right) \nabla_{\mathbf{x}} f(\mathbf{x})$. Then, we have

$$
\begin{aligned}
\nabla_{\mathbf{x}} \mathcal{L}(f_\infty(\mathbf{x}), y) &= \left( \sigma(f_\infty(x)) - \frac{y+1}{2} \right) \nabla_{\mathbf{x}} f_\infty(\mathbf{x}) \\
&= \left( \sigma(f_\infty(x)) - \frac{y+1}{2} \right) \nabla_{\mathbf{x}} \sum_{i=1}^{n} f^{(i)}(\mathbf{x}) \\
&= \left( \sigma(f_\infty(x)) - \frac{y+1}{2} \right) \sum_{i=1}^{n} \nabla_{\mathbf{x}} f^{(i)}(\mathbf{x}) \\
&= \left( \sigma(f_\infty(x)) - \frac{y+1}{2} \right) \sum_{i=1}^{n} \frac{1}{\left( \sigma(f^{(i)}(x)) - \frac{y+1}{2} \right)} \nabla_{\mathbf{x}} \mathcal{L}(f^{(i)}(\mathbf{x}), y) \\
&= \sum_{i=1}^{n} \underbrace{\frac{\left( \sigma(f_\infty(x)) - \frac{y+1}{2} \right)}{\left( \sigma(f^{(i)}(x)) - \frac{y+1}{2} \right)}}_{\alpha_i} \nabla_{\mathbf{x}} \mathcal{L}(f^{(i)}(\mathbf{x}), y).
\end{aligned}
\tag{33}
$$

$\square$

## D.2   Additional Plots

Complementing Fig. 2 in the main text, we show (the first 100) NTK features in Robustness - Usefulness space defined in Sec. 4 for a larger number of architectures for both MNIST and CIFAR in Fig. 2 and 3. We use available analytical NTK expressions for standard FC{1,2,3,4,5} and CONV{1,2} architectures in the NTK regime to evaluate and decompose kernels on a subset of 10K MNIST training images and 10K binary CIFAR images - 5K cars and 5K airplanes. We note that within a dataset, the plots do not change much between architectures, speaking to the universal nature of these kernel-induced features.

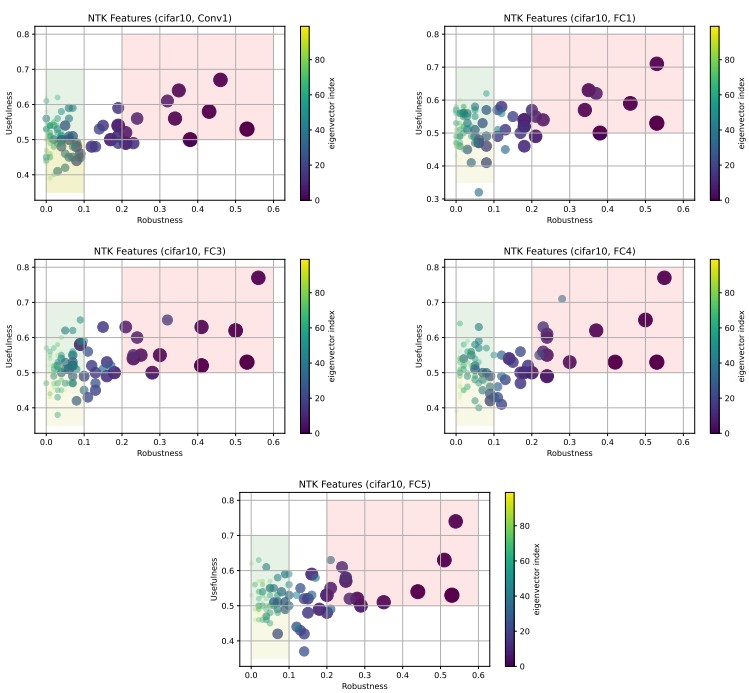

Figure 3: Robustness Usefulness space for various kernels, CIFAR-10 car vs plane. The axes lie in $[0,1]$. Fig. 2 in the main text shows FC2 and CONV 2. "Useful" features have usefulness $> 0.5$ (the random guessing probability for a binary balanced data set). The colored red, green and yellow boxes are arbitrary, meant to visually distinguish useful-robust from other features.

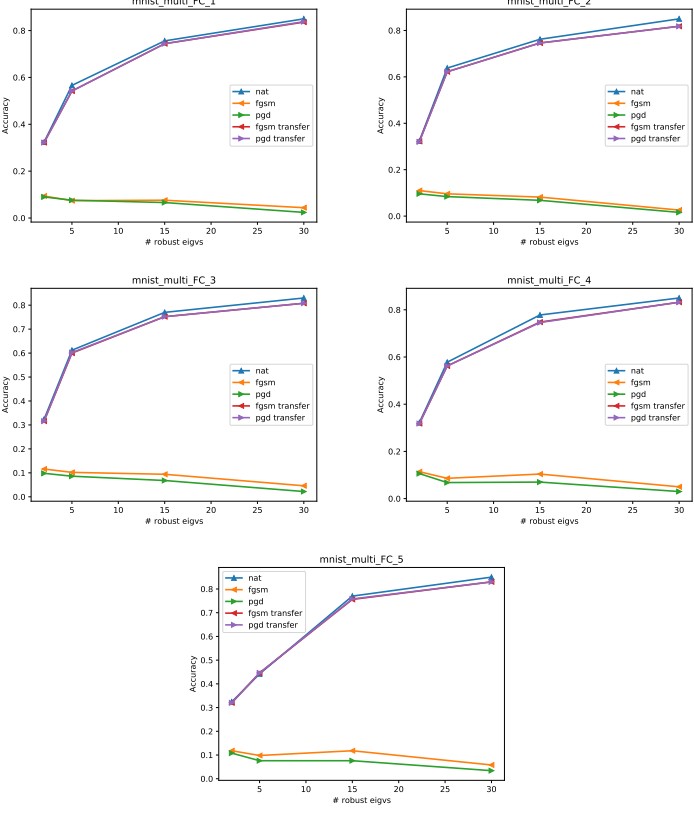

Figure 4: Robustness of kernel when keeping a few of the top robust features (MNIST). The Gram matrix is computed using 10k images from the training set. Blue lines show clean accuracy, red and purple (lines are overlapping) show accuracy against FGSM and PGD10 examples generated using the full kernel machines (consisting of all the features) and orange and green show the resulting robustness of the new model (FGSM and PGD10, respectively). Accuracy on the $y$-axis lies in $[0, 1]$.

### D.3 Robust Features Alone are not Enough

Feature definitions outlined in Sec. 4 open an avenue to use traditional feature selection methods to search for robust models. In particular, here we rank the features of an NTK based on their robustness on a validation set (accuracy against adversarial examples computed from the same feature - setting: FGSM with $\epsilon = 0.3$ for MNIST or $\epsilon = 8/255$ for CIFAR-10). Specifically, we test and rank each "one-feature kernel" function $f^{(i)}(\mathbf{x}) := \lambda_i^{-1}\Theta(\mathbf{x}, \mathcal{X})^\top \mathbf{v}_i \mathbf{v}_i^\top \mathcal{Y}$. Given this ranking, we construct a sequence of new kernels $\Theta'_r(\mathcal{X}, \mathcal{X})$ by progressively aggregating the $r$ most robust features with their original eigenvalues. This gives rise to kernel machines of the form $f'_r(\mathbf{x}) = \Theta(\mathbf{x}, \mathcal{X})^\top \Theta'_r(\mathcal{X}, \mathcal{X})^{-1}\mathcal{Y}$, where $r$ indicates the number of top robust features kept. We present the results of this approach in Figures 4 (MNIST) and 5 (CIFAR-10), where we plot clean accuracy as well as robust accuracy against perturbation from the kernel $f'_r$ itself as well as against "transfer" perturbations from the original (full) kernel.

On the binary classification task, some robustness can be garnered by keeping the most robust features and there seems to exist a sweet spot where the robustness is maximized (this seems to be consistent across other models as well). On multiclass MNIST, however, despite the relative simplicity of the dataset, we are not able to obtain non-trivial performance without compromising robustness. We conclude that it is unlikely that robust features (of standard models) alone are sufficient for robust classification, and the burden of some data augmentation, like in the form of adversarial training, seems necessary, at least for the models considered in our experiments.

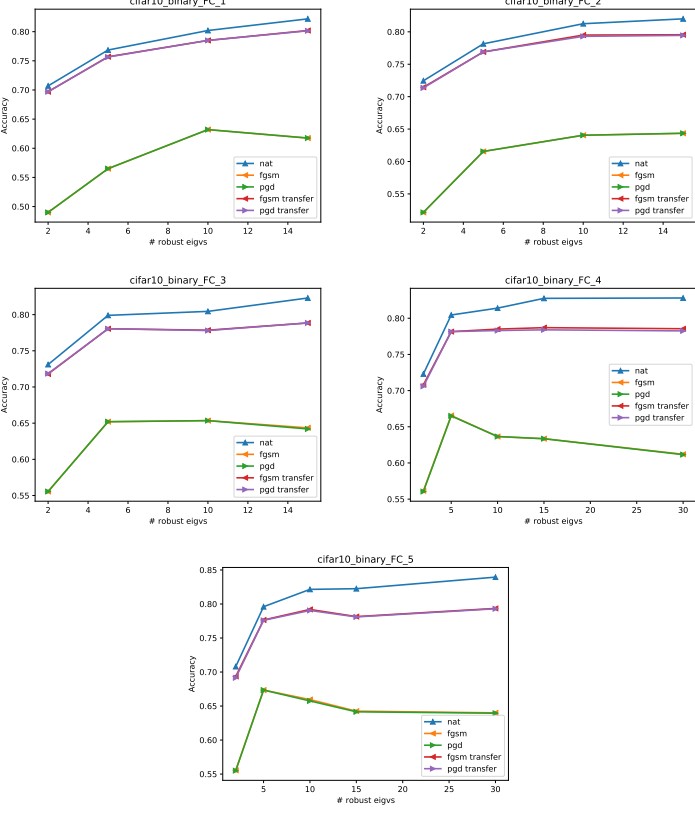

Figure 5: Robustness of kernel keeping a few of the robust features (CIFAR car vs plane). The Gram matrix is computed using all 10K images from the training set. Blue lines show clean accuracy, red and purple (lines are overlapping) show accuracy against FGSM and PGD10 examples generated using the full kernel machines (consisting of all the features) and orange and green show the resulting robustness of the new model (FGSM and PGD10, respectively). Accuracy on the $y$-axis lies in $[0, 1]$.

# E   Experimental Details for the Kernel Dynamics Section

Here we provide the details of our experiments in Sec. 5, where we compare standard and adversarial training by tracking several kernel quantities.

For experiments with MNIST, we use a simple convolutional architecture with 3 layers. The first 2 layers compute a convolution (with a 3×3 kernel), followed by a ReLU and then by an average pooling layer (of kernel size 2×2 and stride 2). The 3rd layer is fully-connected with a ReLU non-linearity, followed by a linear prediction layer with 10 outputs. The layers have width 32, 64 and 256, respectively.

For CIFAR-10, we use a deeper architecture consisting of 6 layers. Layers 1 and 2, 3 and 4, 5 and 6 are fully convolutional with 32, 64 and 128 channels, respectively, and a kernel of size 3×3. There is a max pooling operation after layer 2, and average pooling after the final layer, followed by a linear prediction layer. Both pooling operations use a kernel of size 2×2 and stride 2.

We use a fixed learning rate of $10^{-2}$ for all experiments and no weight decay. We do not use any data augmentation, since we are interested in analyzing the behavior of kernels, rather than obtaining the best possible results. Stochastic gradient descent is used in all cases, with a batch size of 300 for MNIST and 250 for CIFAR-10. The kernels quantities are tracked for the same (first) batch during training. For adversarial training, we either used FGSM or PGD (for generating the adversarial examples) with 20 steps against $\ell_\infty$ adversaries. The maximum perturbation size is set to $\epsilon = 0.3$ and $\epsilon = 8/255$ (for MNIST and CIFAR-10, respectively), and in the case of PGD training we use an attack step size of $\alpha = 0.1$ and $\alpha = 2/255$, respectively. Experiments were run with JAX (Bradbury

et al., 2018), and empirical NTKs were computed using the Neural Tangents Library (Novak et al., 2020). Neural nets were trained using Flax (Heek et al., 2020) and the JaxOpt library (Blondel et al., 2021), adapting code available from the JaxOpt repository. This code snippet was licensed under the Apache License, Version 2.0.

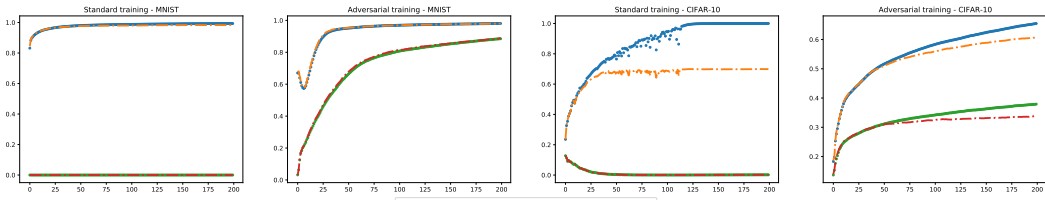

Figure 6: Training curves for networks trained in Sec. 5. From left to right: Standard training on MNIST, Adversarial (PGD-20) training on MNIST, Standard training on CIFAR-10, Adversarial (PGD-20) training on CIFAR-10. For each of the 4 settings, we show train/test accuracy on clean and on adversarially perturbed (PGD-20) data.

Models were trained for 200 epochs. Fig. 6 summarizes the performance of the networks during training. In Fig. 7, we show how norm concentration evolves during training - similar to the plots for CIFAR-10 in Fig. 5, but for MNIST and for two choices of eigenvalue index cut-off.

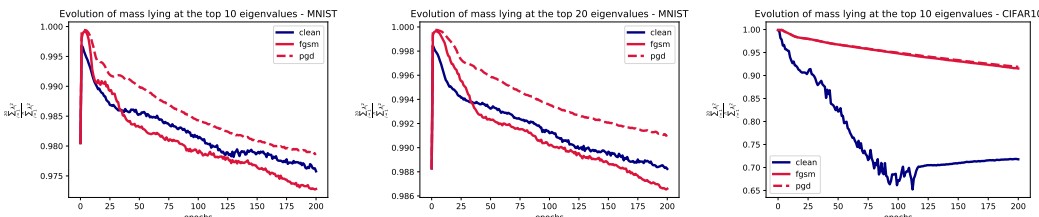

Figure 7: Concentration of norm during standard training vs adversarial training. **Left:** Concentration on top 10 (MNIST). **Middle:** Concentration on top 20 (MNIST). **Right.** Concentration on top 10 (CIFAR-10) (Fig. 5 in Sec. 5 shows Concentration on top 20 for CIFAR-10). For MNIST, we observe that when performing adversarial training with just one-step adversary (FGSM), the mass drops below the level of standard training. This is likely related to a phenomenon called catastrophic overfitting which is widespread in simple FGSM training settings (Wong et al., 2020).

Fig. 8 shows the polar dynamics for the top space (top 20 eigenvalues) of the kernel. We observe little to no change for adversarial training from Fig. 6 in the main text that showed the same information for the entire space, though for standard training there is less rotation in the top space. We entertain this as an indication that adversarial training modifies the "robust" (top) features of the kernel more than standard training.

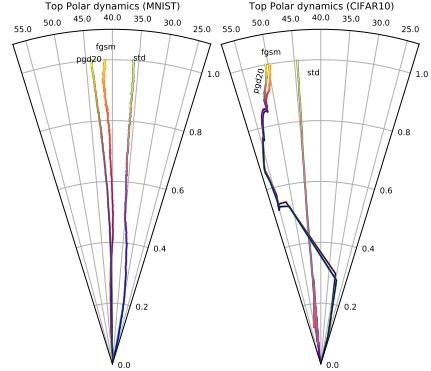

Figure 8: Top-20 dynamics on polar space.

Finally, Fig. 9 shows the values within the kernel matrices before and after training for MNIST for standard and adversarial training. We draw the same conclusions as the main text, namely the "standard" kernel has significantly larger values than the "adversarial" one.

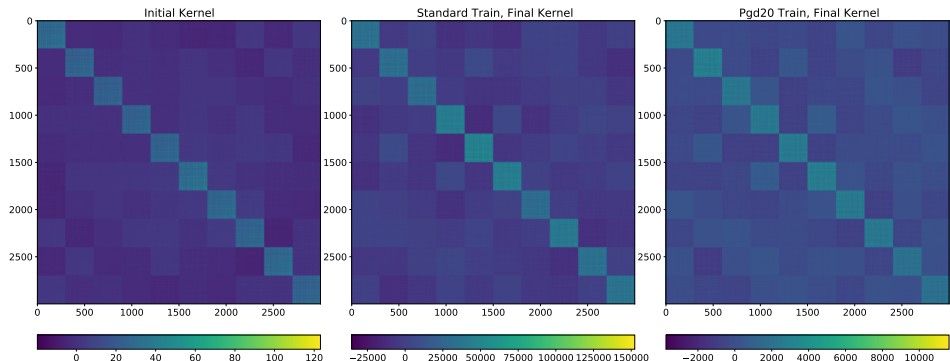

Figure 9: Kernel images for MNIST. Left to Right: Kernel at initialization, Kernel after standard training, Kernel after adversarial training (20 pgd steps). Notice that during training the values increase, but they do substantially more for standard training. Also, observe that for adversarial training there is more spread between different classes. Each little square in the diagonal corresponds to a different class.

### E.1 Linearized Adversarial Training

Motivated by the apparent laziness of the kernel during adversarial training and the findings of prior works (Geiger et al., 2020, Fort et al., 2020) that considered linearization (with respect to the parameters) of the model after some epochs, we do the same for adversarial training.

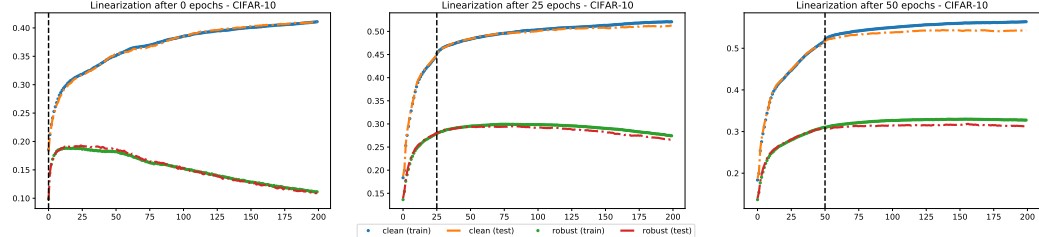

Figure 10: Linearized Adversarial Training on CIFAR-10. **Left** Linearized after 0 epochs. **Middle** Linearized after 25 epochs. **Right** Linearized after 50 epochs. Y-axis has range $(0, 1)$.

We include a small study that linearizes the kernel after a certain number of epochs. In particular, Fig. 10 shows the training behavior after linearizing the CIFAR-10 model after 25 and 50 epochs, and also at initialization. After linearization, we continue adversarial training in this simple linearized model (meaning we generate adversarial examples from the linear model). We observe that adversarial training continues, without a collapse of the training method. In comparison to non-linearized training (Fig. 6), training seems to stagnate. We also observe that the earlier we linearize, the greater the gap is between standard and robust performance. We leave the investigation of this intriguing phenomenon and a detailed comparison to standard training to future work.