# OpenReview forum: "What Can the Neural Tangent Kernel Tell Us About Adversarial Robustness?"
_NeurIPS.cc/2022/Conference — NeurIPS 2022 Accept_

### Official Review · Reviewer_R7n6 · 2022-07-01

**Rating:** 8
**Confidence:** 3
**Soundness:** 3 good
**Presentation:** 4 excellent
**Contribution:** 4 excellent

**Summary:**

The paper applies the theory of Neural Tangent Kernel to explain insightful properties of adversarial examples.

**Questions:**

I would like to clarify the following minor comments
- line 68: "only require ... architecture" and training data as well, isn't it?
- line 239: "NTK ... suggest that these two scenarios are no so distinct" The scenarios are very distinct, the adversarial perturbations as well, the required average distortion for hacking images is bigger in BB than in WB. Everything is different!
- line 243: "NTK as the most faithful BB attack." I disagree. The NTK is another scenario where the attacker has only the architecture and the training data, different from WB scenario where it knows the weights and architecture, different from BB scenario where it knows nothing but has a free access.
- Fig. 2 Right. The colored boxes are not explained in the text. I suppose it shows areas of (non)useful/(non)robust features. How did you set the boundaries of these boxes?


**Limitations:**

Very good submission I enjoy to read.
I dislike paragraph L241-245 that I would remove, freeing some space for more explanation on the computations of the "eigen-images" for instance.

**Strengths And Weaknesses:**

# Strengths
- Very well written, crystal clear, both text and maths.
- Very timely since NTK gains more and more momentum
- Opens the doors to many other discoveries (both about attack and defenses)

# Weaknesses
- Too few details about the computation of the NTK (I was expecting these in the appendix) and its cost.
- Very few statements deserve some mitigation (see below)

---

> ### Author Response · Authors · 2022-08-02
> **Responses to questions, incorporating suggestions**
>
> Thank you for your time and your enthusiastic review! To address your comments:
>  * We will provide slightly more details on the NTK computation in the appendix of the revised version. In Sec. 3.2 we used Eq. (C.29) and its derivative, evaluated on the entire training data. For the robustness-usefulness analysis in Sec. 4 we use the (differentiable) analytical expressions for the various architectures, available in the neural tangents library, and compute kernels of size 10K for MNIST and the entire binary CIFAR (also 10K) - these details were in a caption before and are now in the main text. In general it is harder to compute analytical kernels in the multi-class case, especially for more sophisticated architectures. The details on the empirical kernel dynamics computations in Sec. 5 are in the Appendix.
>  * line 68: You are right that training data are needed. Our intent in this line is to compare the method we present with attacks that train substitute models, where the dependence on data is obvious. So in that sense, there is no change in the assumptions of the two models, which is why we do not mention it there.
> * We appreciate your concerns about paragraph L241-L245. We will modify/shorten these lines in the revision, to make the distinction between our model and the traditional BB and WB models more crisp. We will mention that our threat model explicitly requires access to the training data, and how this may depart from the currently established definition of the black-box threat model. We will make similar clarifications in the other places you point out. Thank you for alerting us to this potential source for confusion: we will disambiguate our claims (and remove allusions to the black box threat model in the NTK context).
> * Concerning Fig. 2, you are right. The colored boxes are for illustrative purposes and the boundaries are chosen arbitrarily to help distinguish the features. We have added a note in the caption of the revised version.
>
> Thank you for these valuable suggestions.

---

### Official Review · Reviewer_vnuA · 2022-07-09

**Rating:** 4
**Confidence:** 3
**Soundness:** 2 fair
**Presentation:** 2 fair
**Contribution:** 3 good

**Summary:**

This work generates adversarial examples by using a training-free fashion through NTK. They study the feature through the eigendecomposition of the associated kernel, and confirm that the robust features are corresponding to the top part of the eigenspectrum. They also study the dynamics of empirical NTK during adversarial training.

**Questions:**

1. It’s unclear what’s the architecture of the neural network the author describes in section 2.3, which should be given math formulas. Does the author focus on two-layer ReLU network since it dominates NTK literature? It should be clear whether the weight w a vector or a matrix; when training the neural network, whether any part of the layer is fixed; how to initialize the neural network The above questions are different throughout different NTK literature.
2. The author presents the eq(7) for learning with L2 loss, which is described by a regression problem. Then try to transfer to the classification problem by simply applying the softmax function on the top of the kernel output and applying cross-entropy loss. I’m wondering whether the author considers the training dynamics of learning with cross-entropy loss for infinite neural networks.
3. Given eq(8), does the author also consider any multiple-step adversarial example for the infinitely wide neural network?
4. The author should describe more on ``kernel generated attack’’, i.e. how to generate adversarial examples from kernel machines. Since the NTK is solely based on model initialization, does the attack only based on one random initialization or an average of multiple initializations?
5. For the top figure of figure 4, as you increase the epoch, the cosine similarity between NTK and neural network is decreasing. Moreover, it should be helpful if also plots how far the network weight is away from the initialization during training. If the learning rate is very small, then even after 10^5 epoch, there’s possible that the trained neural network is still not far from the initialization, thus lying within the lazy regime. So we should not be surprised by the result. And the author should do a thorough test of how these attacks perform compared with white-box attacks.
6. Mathematically it's not correct to use $L(f-f_t,y)$ in equation (C.30). In figure C.1 (a), the bottom figure is about epoch vs. robust accuracy, and why the highest accuracy can reach 200%?
7. Some of the figures need to enlarge the text size, i.e. figure 3, 6.

**Limitations:**

Yes.

**Strengths And Weaknesses:**

Strength:

1. I find the idea of connecting robust features with the eigenspectrum of NTK and studying the kernel dynamic of standard training vs adversarial training is novel. Some of the experiments are interesting.

Weakness:
1. The experiment cannot convince me that the black-box attack through NTK has a similar effect as while box attack. The strong alignment of the gradient is likely because the trained neural network is within the lazy regime, yet in reality, the neural network cannot guarantee to be within the lazy regime, i.e. adversarial training, thus making the proposed black-box attack not useful.
2. The author does not describe the technical detail clearly in the main paper. There are lots of NTK literature under different model architectures, different weight initialization, different data distribution assumptions, etc. Yet I did not find the author considering all the above situations thoroughly. Later, I notice that the author describes the initialization in appendix C, where they consider two-layer ReLU network with $W\sim N(0,0.01^2I_{m\times d})$. To me, such initialization is neither practical nor falls into the standard NTK literature.
3. I have trouble understanding the reason for ``robust features alone is not enough''. The author defers the discussion in the appendix, where CIFAR10 binary FC_4 FC_5 suggests that adding more features may hurt the robust performance. The author however compares it with multi-classification MNIST and concludes that robust features are not sufficient for robust classification. First MNIST is a rather simple unique dataset and it's sparse. It's likely that there're so few robust features for MNIST. Second, it's unfair to compare a binary classification result vs a multi-classification result on different datasets. At least present the same plot for binary MNIST and multi-class CIFAR10.

---

> ### Author Response · Authors · 2022-08-02
> **Responses to reviewer questions and clarifications**
>
> Thank you for your time and your review! We address your concerns:
>
> * We agree that our claim is valid only in the lazy regime. This is reflected in the title of this section "White box = Black box in the kernel regime" (which we have now adjusted). We realize that this generated confusion, and will make sure to delineate the claim clearly in the rest of the section.
> * We considered the same setup as in Arora et. al, ("Fine-Grained Analysis of Optimization and Generalization for Overparameterized Two-Layer Neural Networks", ICML'19) which is one of the first works in the NTK literature.
> * We apologize if that section in the supplementary material was not clear enough. We will try to clarify the confusion in the revised version. The point is that we do not compare performance on binary problems with performance on multiclass, we just argue that achieving robustness on multiclass classification is generally more difficult than in the binary case (see for instance: Qian et al., "Robustness from Simple Classifiers", https://arxiv.org/abs/2002.09422, 2020). Based on the results on MNIST (which did consider a very small amount of robust features - 3), we conclude that it is unlikely that a kernel composed of robust features will yield a robust machine.
>
> Regarding your questions:
>
> 1. We review aspects of the NTK theory in the most general setting, while referring interested readers to the relevant papers for the exact assumptions of the theory. We clarified the confusion around the dimension of the weights in the revised version. Thank you for suggesting this. As indicated in the Appendix, for the transfer results in Sec. 3.2 we use the two-layer network with frozen weights of Eq. (C.29). For the usefulness-robustness results we use standard FC{1,2,3,4,5} and CONV{1,2} architectures, which analytical expressions provided in the neural tangents library.
> 2. Anlyzing the minimization of cross entropy loss in the lazy regime is a non-trivial subject, where the neural net expressions change significantly. See for example [Lee et al 19] Appendix B.2. We do not attempt to pursue this direction in our paper.
> 3. Multi-step attacks are harder to be analyzed analytically, so we only consider this scenario in the last section, where we do adversarial training.
> 4. Appendix B contains the derivation of the attacks on kernels. Please let us know if you have any suggestions on how to improve it.
> 5. In that section, we deliberately stay in the kernel regime, as reflected in the title of the section. We will make sure to make it more clear in the revised version.
> 6. Please note that the accuracy reaches 2% and not 200% (the y-axis shows %).
> 7. Thank you for the suggestion! We will improve the visibility of figures in the revised version!
>
> In summary, we will make the setting of Section 3.2 more clear, as you suggested and as is suggested by some of the other referees. Note, however, that the role of that Section is mostly to motivate the analysis that follows (and our revised version will adjust the claims in the introduction that refer to it). We are at your disposal for further questions about the rest of the paper and hope that you might consider raising your score, since you seem to appreciate its contributions.

---

> > ### Comment · Reviewer_vnuA · 2022-08-06
> > **Response to authors**
> >
> > I thank the author for the detailed response and make efforts to modify the manuscript. I have some following arguments regarding the weakness part.
> > 1. The transfer learning attack only holds for the lazy regime, then if the network is outside the lazy regime, the black box attack from this paper will not have the same effect as white-box attack, therefore my first weakness still holds.
> > 2. For [1], the initialization is $W\sim(0,\kappa^2I)$, and as far as I can tell, $\kappa$ is for the purpose of analysis to understand how width changes with $\kappa$. In fact, [2,3,4,5] all use the same standard initialization $W\sim N(0, I)$ with $1/\sqrt{m}$ normalization factor. Therefore, it’s still unclear to me why choosing $\kappa=0.01$.
> > 3. I appreciate the authors' thoughts on different experiments. However, since this is more of an experimental paper, merely binary classification with two-layer ReLU network won't make a strong argument or any insightful suggestions for real-world application. Does the author try any transfer-attack experiments for multi-layer neural networks? It’d be more convincing if the idea could go beyond two-layer networks.
> >
> >
> > [1] Fine-Grained Analysis of Optimization and Generalization for Overparameterized Two-Layer Neural Networks
> >
> > [2] Ji, Ziwei, and Matus Telgarsky. "Polylogarithmic width suffices for gradient descent to achieve arbitrarily small test error with shallow relu networks." arXiv preprint arXiv:1909.12292 (2019).
> >
> > [3] Du, Simon S., et al. "Gradient descent provably optimizes over-parameterized neural networks." arXiv preprint arXiv:1810.02054 (2018).
> >
> > [4] Cao, Yuan, and Quanquan Gu. "Generalization bounds of stochastic gradient descent for wide and deep neural networks." Advances in neural information processing systems 32 (2019).
> >
> > [5] Cao, Yuan, and Quanquan Gu. "Generalization error bounds of gradient descent for learning over-parameterized deep relu networks." Proceedings of the AAAI Conference on Artificial Intelligence. Vol. 34. No. 04. 2020.

---

> > > ### Author Response · Authors · 2022-08-07
> > > **Response to comments, parameter selection and pointing out multi-class experiments**
> > >
> > > We thank the reviewer for their response to our comments, and appreciation of our efforts. Perhaps the following clarifications can address your comments, and in particular **convey our complete surprise to read your comment/weakness 3 since we made special effort to include many multi-class experiments**.
> > >
> > > > 1. The transfer learning attack only holds for the lazy regime, then if the network is outside the lazy regime, the black box attack from this paper will not have the same effect as white-box attack, therefore my first weakness still holds.
> > >
> > > You are correct that in Section 3 we present transfer results to wide nets in the NTK parameter setting. Our main goal here is to make sure gradients (with respect to the data) transfer from NTKs to wide nets as we would expect (but which has not been verified anywhere in the literature, to our knowledge). We then discuss the type of threat model such an "NTK-attack" would constitute (it requires description of the model architecture and training data, but not more, thus is somewhat outside the standard attack models that are usually studied). Having established these two facts, we then open the question for future work (see lines 369-371: *"Sec. 3 argues that transferable attacks from the NTK **may be** as effective as white-box attacks, but this warrants an in-depth study across architectures, kernels and data sets (which has not been the main focus of this work)."*) In this paper, we aim to lay out a wider spectrum of benefits the "NTK-lens" offers to the study of adversarial robustness: this is not intended as a cryptography paper proposing a new attack. We truly hope the reviewer can appreciate our other contributions as well (Sec. 4, 5 and the analytical derivations complimenting Sec 3) and want to convey the spirit of this paper as one that opens the route to wider inquiry.
> > >
> > >
> > > > 2. For [1], the initialization is $W\sim(0,\kappa^2$ I), and as far as I can tell,  is for the purpose of analysis to understand how width changes with $\kappa$. In fact, [2,3,4,5] all use the same standard initialization $W\sim N(0, I)$ with $1/\sqrt{m}$ normalization factor. Therefore, it’s still unclear to me why choosing $\kappa=0.01$.
> > >
> > >
> > > We appreciate your concern and thank you for the additional references, which are all important works. As already mentioned in our previous comment, we have decided to consider the same setup as in Arora et. al, ("Fine-Grained Analysis of Optimization and Generalization for Overparameterized Two-Layer Neural Networks", ICML'19) which is one of the first (and arguably influential) works in the NTK literature (reference [1] from your list). Please allow us to cite from the arXiv version of that paper (https://arxiv.org/pdf/1901.08584.pdf): page 17, "Experiment Setup", first paragraph:
> > >
> > > * "...Our theory requires a small scaling factor κ during the initialization (cf. (1)). We fix $κ = 10^{−2}$ in all experiments...."
> > >
> > > Note that Arora et al. have the $1/\sqrt{m}$ term in their setup (formula above Eq. (1) on page 1) - as we do in ours (Eq. (C.28)). While Du et al. (reference [2]) consider only the case $\kappa = 1$ (see footnote 3 on page 5 of [1]) [1] includes $\kappa$ also into bounds in $m$ and, relevant for us, sets $\kappa$ to a small constant in the experiments. As far as we could tell, it is important that smaller $\kappa$ results in smaller outputs at initialization. (Perhaps related is a similar parameter in [Arora et al. '19: On Exact Computation with an Infinitely Wide Neural Net] that multiplies the output of the net.)
> > > We hope this clarifies our choices, but we are at your disposal for further clarifications, if needed.
> > >
> > > > 3. I appreciate the authors' thoughts on different experiments. However, since this is more of an experimental paper, merely binary classification with two-layer ReLU network won't make a strong argument or any insightful suggestions for real-world application. Does the author try any transfer-attack experiments for multi-layer neural networks? It’d be more convincing if the idea could go beyond two-layer networks.
> > >
> > > We are truly surprised at this comment. Our paper contains non-binary (multi-class) experiments for each topic we touch upon (transfer to wide nets, features and their visualization, kernel dynamics). Might the reviewer perhaps have overlooked Appendix C (which is entirely dedicated to transfer in the multi-class case) and reference to it in lines 232-234 (of the revised version of Aug 3) - *"We reproduce these plots for MNIST in the Appendix, leading to similar conclusions."*?  While space constraints didn't allow us present *all* multi-class experiments in the main body of the paper (they are in App. C and D) we have all of Section 5 completely in the multi-class setting.
> > >
> > > We hope our responses are satisfactory and might incite you to reconsider your score, since you seem to appreciate the novel connection between the NTK and adversarial robustness.

---

### Official Review · Reviewer_bexg · 2022-07-09

**Rating:** 6
**Confidence:** 5
**Soundness:** 3 good
**Presentation:** 3 good
**Contribution:** 3 good

**Summary:**

This paper summarizes a novel empirical study on the adversarial robustness of neural networks using a neural tangent kernel perspective (both for finite-width networks and their infinite-width limit). The main new insights of this work are:
1. The adversarial perturbations of a trained infinite-width NTK match quite significantly those of the same wide-but-finite architecture trained on the same dataset.
2. For the NTK at initialization, the first eigenfunctions of the kernel tend to correspond to robust and useful features, while latter eigenfunctions mainly correspond to non-robust, but also useful features. The best performance is achieved by leveraging both types of features, and robustness can not be obtained by training only on the subspace of robust features at initialization.
3. The evolution of the finite-width NTK is much faster for adversarial training than standard training. The authors argue that this evolution is fundamental to yield robustness through feature learning.

**Questions:**

I would appreciate if the authors could answer my doubts with regards to the technical details that I outlined in Weaknesses 1. I am afraid that these could be hints of potential technical flaws in this work, and hence they are are the main reasons why I am slightly leaning towards rejection. However, If the authors can alleviate my concerns, I will be very happy to significantly increase my score, as I believe this paper could be a nice contribution.


**(*Minor*) Fix citations**: I have spotted that several references point out to arXiv papers that have already been accepted to multiplee archival venues. I would encourage the authors to review their citations to make sure they correctly cite each article.

**Limitations:**

I see no direct negative societal impact stemming from this work.

**Strengths And Weaknesses:**

## Strengths

1. **Timely and interesting perspective**: The NTK perspective has been the object of intense scrutiny by the generalization community in the past few years. In this regard, this work nicely complements these prior studies by filling a gap in the literature analyzing through the NTK lense different phenomena related to adversarial robustness. This will not only be useful for the robustness community, but also for the deep learning theory community at large, since studying adversarial training through this lense can give complementary information of the general dynamics of deep learning, as this is a very different algorithm. Personally, I find the fact that there are qualitative similarities in the evolution of the NTK between adversarial training and standard training, but also significant quantitative differences in terms of speed and magnitude of the change, quite remarkable.

I also find the study of the robustness of different eigenmodes of the NTK at initialization, and the transferability of adversarial attacks between NTKs and NNs to be very interesting. However, as I point out in the weaknesses, I think there are some fuzzy technical details in these experiments that I would like the authors to clarify.

2. **Balanced and self-contained analysis**: In general, I found this paper very easy to read and quite balanced. The authors touch upon different topics with the right amount of breadth and depth that gives the reader a good amount of new information.

3. **Good contextualization with prior work**: The authors properly discuss a great deal of relevant work, both in the space of deep learning theory and adversarial robustness. I believe this will makes this paper easier to follow by both communities.

4. **Opens new research avenues**: As a new empirical study on relevant deep learning phenomena, I significantly appreciate that this work gives a clear path forward for future theoretical and practical studies. In particular, I believe that the analysis of the after-kernel of adversarially trained networks could yield significant insights for robustness, while the study of the accelerating effect on the kernel evolution of adversarial training could also help understand the rich regime of deep learning as a whole.

## Weaknesses

1. **Fuzzy technical details**: I find some technical details of this work a bit hard to follow and somewhat suspicious of possible flaws in the evaluation. Specifically, I am referring to:
- **Number of epochs of training in Fig. 4 and corresponding robustness**: The x-axis in Fig. 4 is in logarithmic scale and shows the evolution of the robustness of a neural network 100,000 epochs! This is (i) an oddly high number, and (ii) surprising for the fact that the robust accuracy is very high at the "beginning of training" (100 epochs). This does not match the previously reported evolution of the robustness of neural networks in standard training, and hence makes me seriously doubt of the validity of this experiment.
- **Unclear usefulness metric**: I find unclear which precise metrics are used in Fig. 2 (right). In sec. 2.2. the authors introduce two metrics for usefulness and robustness in Eq. (3) and Eq. (4) but then later mention that they use clean accuracy and robust accuracy of the different eigenmodes as metrics. Knowing which of these two alternatives the authors use to plot Fig. 2 (right) is very important to contextualize the reported numbers.

2. **Weak results on transferability/black-box attacks**: Although I understand the reasons behind selling their transferability results as a black-box attack, I honestly believe that the provided evaluation (even despite my previous comments) is not enough to claim this technique as a useful adversarial black-box attack. The context of this experiment is rather limited, and has only studied the transferability between very wide networks and the NTK. These two models have been shown before to share very similar dynamics, and in this regard, it is not surprising that their adversarial attacks transfer between each other. This section would have been much stronger if it had also studied the transferability of adversarial attacks between linearized neural networks (i.e, finite-width NTK) and standard neural networks. If the transferability is high in that setting as well, then claiming this as a promising black-box attack technique would not be overclaiming.

3. **(*Minor*) Figure design**: Some graphic design aspects of the figures could be improved to help readability of the paper:
- The colormap in Fig. 3 (left) is far from ideal. The current figure is using a qualitative colormap to show a numerically diverging quantity. I would advice the authors to change it some [perceptually uniform and sequential colormap](https://matplotlib.org/stable/tutorials/colors/colormaps.html).
- The font size is very small. It is in general very hard to read the labels and titles of all the figures.

---

> ### Author Response · Authors · 2022-08-02
> **Responses and clarification of technical details, and thanks!**
>
> Thank you very much for your comprehensive and valuable review. We have been thrilled to read your very thoughtful comments and the enthusiasm you convey for our study and are very grateful for your remarks, which we have taken as a starting point for a few improvements of our paper. We hope that the answer to your questions and changes we have made alleviate your concerns as to possibly fuzzy details and technical flaws. We truly hope our clarifications will incite you to raise your score, and are at your disposal for further clarifications if necessary.
>
> To answer your questions:
> > 1. Number of epochs of training in Fig. 4 and corresponding robustness:
>
> * You are right that these numbers would look suspicious under "normal" conditions. However, note that these experiments were performed in a different regime ("close to the NTK"): The number of epochs is high, since learning is slowed down due to the choice of hyperparameters (learning rate, variance at initialization, large width, $\ell_2$ loss, full batch GD - please see Appendix C for details). The robustness seems very large, but notice that Fig. 4 shows a binary task and hence the numbers must be appreciated accordingly. The same experiment, when done on multiclass MNIST (Fig C.1(a)), shows very low levels of robustness (1%), which agrees with previous literature (which mainly focuses on multiclass problems). Finaly, notice that robustness being larger in the beginning than in later stages of training agrees with our finding that the top eigenfunctions of the NTK are more robust (as the top ones are the ones being learned first when nets are trained in the kernel regime).
>
> >2. Unclear usefulness metric:
>
> * Good catch - thank you! Indeed, our definition of usefulness and robustness of a feature differs slightly from what we present in Fig 2 (right) and Figs. D.2 and D.3. We corrected this in the revised version. In brief, we view the usefulness of a feature as its classification ability (and, accordingly, its robustness as its classification ability under adversarial perturbations). This makes the extension to the multiclass case natural (App. A). In our figures, we use robustness against FGSM attacks as a surrogate for robustness, as mentioned in Sec. 4.
>
> >3. Weak results on transferability/black-box attacks
>
> * You are absolutley right that our experimental analysis in Section 3.2 is limited to the kernel regime (and we do mention this throught the paper, though possibly not enough). In the revised version, we are making sure to put  the contributions of this section in a better context. Doing the same set of experiments with empirical kernels and more realistic networks (such as deep convolutional architectures) is definitely interesting and important, albeit computationally demanding and sligthly beyond the scope of this paper in our opinion. We will try to emphasize that the contributions of this section serve more as a motivation for the remainder of the paper, and adjust the claims about black-box attacks.
>
> > Miscellanea: Figure colors, font sizes, references:
>
> * Thank you for pointing these out. We have modified the color map of Fig. 3 to something hopefully more satisfying. We updated all references, and increased the font size where possible.
>
> Thank you for helping us improve our work! We hope our responses address your concerns.

---

> > ### Comment · Reviewer_bexg · 2022-08-06
> > **Thank you for the improvements to the manuscript**
> >
> > Thank you very much for your answer and for the effort in improving the paper. I honestly believe the current version of the manuscript is more precise and polish. I also appreciate the clarifications with regards to the training regime setup and the reasons for the high robustness in Fig. 4. Based on your new comments I now think that the reported numbers do indeed make sense.
> >
> > I will increase my score to 6: Weak accept, as I believe the paper has merit. Overall, I believe this paper provides a nice initial study on the connections between NTK theory and adversarial robustness and deserves to be accepted. It is true, as some other reviewers mention, that some experiments could have been extended further or be a bit more thorough (e.g., evaluate with PGD, rather than FGSM; or performed thorough comparisons for practically-sized finite-width linearised networks), but I do not think these are enough reasons to argue for a rejection.
> >
> > However, on this last point, I would like to add that, to the best of my knowledge, it is in fact not prohibitive to compare the dynamics of a standard neural network and its linearised counterpart. This has been previously done by (Fort et al. 2020, Baratin et al. 2021, Ortiz-jimenez et al. 2021) without access to much hardware. For the same reason, computing adversarial attacks of the linearised neural networks should not require a prohibitive computational cost if performed using standard gradient-based attack pipelines.
> >
> > - Fort et al. Deep learning versus kernel learning: an empirical study of loss landscape geometry and the time evolution of the neural tangent kernel. NeurIPS 2020.
> > - Baratin et al. Implicit regularization via neural feature alignment. AISTATS 2021
> > - Ortiz-Jimenez et al. What can linearised neural networks actually say about generalisation? NeurIPS 2021

---

### Official Review · Reviewer_Ly8J · 2022-07-12

**Rating:** 4
**Confidence:** 4
**Soundness:** 2 fair
**Presentation:** 2 fair
**Contribution:** 2 fair

**Summary:**

This work uses the Neural Tangent Kernel (NTK) as a tool to study several aspects of adversarial robustness.
First, it uses the NTK to generate adversarial examples and find they transfer to the trained counterparts.
Second, it uses the eigendecomposition of the NTK to define features, which correspond to each eigenvector of the kernel matrix on the training data. It analyzes these features by visualizing them and evaluating their usefulness and robustness.
Lastly, it analyzes the evolution empirical NTK for standard and adversarial training. In each of these experiments, it contrasts its findings with the existing literature on adversarial robustness.

**Questions:**

- Have you tried visualizing the features by looking at the most/least activating images? Gradients (as visualized in the paper) are useful, but it's common to look at a few other visualizations
- Can you provide more details on how exactly you computed usefulness and robustness for the NTK features? I could not find them in the Appendix.
- Have you tried computing the robustness of the eigen-features arising from the empirical NTK?
- For the polar angle plots, can you add a legend for the color (epoch?)?

**Limitations:**

None other than those mentioned above.

**Strengths And Weaknesses:**

Strengths
- Using the NTK to study adversarial robustness is a new contribution (I was surprised there were no prior works, except for some overlap).
- Each of the experiments are quite interesting on their own, and some of them appear to be novel. I particularly liked the eigendecomposition of features.

Weaknesses
- Overall, the main issue is that none of the experiments quite go deep enough, and overall I'm not sure what I learned from the paper. Authors argue that their results unify all the known existing phenomena. Surely, while all of their results are *consistent* with what is known in adversarial robustness and NTK, I'm not sure there was something new learned.
More details:
  - The adversarial transfer experiment is interesting, but pretty much expected given earlier works that empirically evaluated to the similarity of NTK predictions and their finite-width counterparts (e.g., https://arxiv.org/abs/1912.02803). There also appears to be prior (unpublished) work on this exact experiment: https://openreview.net/forum?id=M5hiCgL7qt. What would have been interesting to study is how transferability varies as your training regime deviates from the NTK regime; without this study, I feel that the claims about the "black box" nature of this attack is a bit overclaimed.
  - The third investigation (evolution of empirical NTK) seems to make a couple of leap of faiths (maybe they are better justified but in that case you should make them clearer). In particular, it is not obvious at all why the top eigenspectrum should correspond to more robust features for the empirical NTK and particularly for adv training (as this was only shown for the exact NTK and std training). I wonder why the authors did not try computing this explicitly (see Questions).
  - I liked the E.1 Linearized Adversarial Training experiment, but wish the authors had developed this further.
- Second issue I have with the paper is the style of writing. It is polished, but sentences are overly long/flamboyant and not to the point.

Despite these drawbacks, I appreciate the authors' breath of empirical investigations (including those in the app).
Overall, the paper would benefit from having a more focused thesis/hypothesis and tailoring the experiments to that thesis.

---

> ### Author Response · Authors · 2022-08-02
> **Response to reviewer's comments and questions, clarifications**
>
> Thank you for your time and your review! We appreciate your critical read of our work. Before we adress your questions, please allow us to comment on the intent of this paper: In general, since adversarial robustness has not been studied much (or at all) from an NTK viewpoint, we chose breadth over depth in many places of our work, also trying to open new areas of exploration for future work. However, the fact that adversarial examples transfer (in the kernel regime), the fact that the distinction between robust and non-robust features seems to hold on kernels as well - and that robust features tend to correspond to the top of the eigenspectrum, and the empirical phenomena of movement and laziness of the empirical kernel during adversarial training are all novel contributions of our study. In our revision, we tried to improve the clarity of our claims regarding the so-called black box attack we introduce, as we realize they have been confusing for several of the referees. We hope you will appreciate what we believe is a more nuanced presentation (and hence less flamboyant).
>
> We also agree that our transfer results to wide neural nets are in no way surprising (and were not meant to be): since previous works have not studied or emphasized transfer of *gradients* with respect to the data, we wanted to establish and highlight these results chiefly as a basis for what follows and to make sure the foundations hold (i.e. gradients behave as we expect).
>
> >I liked the E.1 Linearized Adversarial Training experiment, but wish the authors had developed this further.
>
> Thank you! We were severely constrained by space and scope, and have left extensions of this particular line of research to future work. However, in the revision, we have added an additional set of data comparing linearization at initialization (previously we only linearized after 25 and 50 epochs) and expand our observations on the resulting gap between standard and robust test accuracy (which increases the earlier we linearize). We believe this phenomenon warrants further in-depth study.
>
> In reponse to your questions:
>
> > The third investigation (evolution of empirical NTK) seems to make a couple of leap of faiths (maybe they are better justified but in that case you should make them clearer). In particular, it is not obvious at all why the top eigenspectrum should correspond to more robust features for the empirical NTK and particularly for adv training (as this was only shown for the exact NTK and std training).
>
> We agree that this later part of our discussion (L. 351-359) is speculative and makes the assumptions  you point out (a leap of faith from the analytical to the empirical NTK). We only attempt to provide some intuition on the mechanism of adversarial training through what has been analyzed already in the paper. However, we believe that the rest of this section makes interesting contributions (e.g. slowdown of kernel during adversarial training) that are not based on any assumptions.
>
> > Can you provide more details on how exactly you computed usefulness and robustness for the NTK features? I could not find them in the Appendix.
>
> * Thank you for asking this! The way that we computed them indeed differs slightly from the definitions of Section 2.2. The revised version corrects this. In brief, we view the usefulness of a feature as its classification ability (and, accordingly, its robustness as its classification ability under adversarial perturbations). Please let us know if you have any more questions about this.
>
> > Have you tried computing the robustness of the eigen-features arising from the empirical NTK?
>
> * This is a very good suggestion and we were initially planning to include such a study in our paper. However, computing the empirical kernel for convolutional, multi-output architectures for the **whole dataset** and then computing its gradients to estimate robustness is computationally prohibitive at the moment (at least to the best our knowledge). So, reproducing the study of Section 4 for empirical kernels is not trivial computationally. We believe that this is one of the possible future directions that our work opens.
>
> Please also note that any prior work you might be referring to is unpublished and has (not coincidentally) not been made available on the arXiv.

---

### Author Response · Authors · 2022-08-03
**Summary of changes in the revised version**

Main paper:
1. We have clarified that our transfer results from (analytical) kernels to neural nets are only provided in the "lazy" regime and have clarified statements comparing the "NTK-attack" to black box and white box thread models. We clarify that our claims to the effectiveness of this attack are only verified for the wide networks in the "lazy"-regime considered in Sec. 3.2.
2. We have clarified the nature of the "NTK thread model": it requires training data and knowledge of the model architecture, but no access to model weights, so constitutes a type of "analytical substitution attack".
3. We have slightly modified the definitions of robust and useful features in Section 2.2 and Appendix A according to their classification ability. This gives a straightforward extension to the multiclass case.
4. We have clarified "usefulness" in the captions of Figs. 2 (Right) and D.2 and D.3 in the Appendix and pointed out that the shaded boxes only serve to visualize various regions in usefulness-robustness space
5. We have improved the color rendering of Fig. 3 and increased font sizes where possible
6. We fixed the legend of Fig. 6 Left (which already had a correct rendition in the former appendix, which we now moved to the main text)
7. We have updated all references to refer to the latest published versions

Appendix:
1. In Sec. E.1.we have expanded our analysis of linearized models and added an additional set of data comparing linearization at *initialization* (previously we only linearized after 25 and 50 epochs) and expand our observations on the resulting gap between standard and robust test accuracy (which increases the earlier we linearize). We believe this phenomenon warrants further in-depth study.
2. We slightly modify the definition of usefulness/robustness in App A, as explained for the main paper
3. We have added more detail on the size of the datasets used in Sec. 3.2 (the entire dataset)
4. We explanded the captions on Figs. D.2 and D.3 to indicate y-axis values for useful features and clarifying that shaded boxes serve for visualization only.
5. We clarify in Sec. D.2. the exact architectures and data set sizes we used for the usefulness-robustness calculations

---

### Meta-Review · Area_Chair_e65v · 2022-09-01

**Recommendation:** Accept
**Confidence:** Less certain

**Metareview:**

This paper is a borderline case. The review scores are quite widely spread with 4, 4, 6, 8. More specifically:

- The score 8 review is unfortunately rather short and could be more informative.
- The score 6 review argues that the paper is a nice initial investigation in this direction and would prefer accepting the paper. But they also agree that it's a borderline case and wouldn't argue strongly against rejecting the paper.
- The score 4 reviewers both responded to the rebuttals and decided to maintain their scores. Both of them found the experiments too superficial and would have preferred a more in-depth investigation.

As a result, the paper presents a difficult decision because there is nothing technically wrong about the paper, but the investigation also appears preliminary and lacking depth. My senior area chair suggested to accept the paper, so I'm going with this recommendation. I would not argue against rejecting the paper if the program chairs would prefer this outcome.

**Award:**

No

---

### Decision · Program_Chairs · 2022-09-14

Accept